# Alternative splicing liberates a cryptic cytoplasmic isoform of mitochondrial MECR that antagonizes influenza virus

Steven F. Baker[1,¤a]*, Helene Meistermann[2], Manuel Tzouros[2], Aaron Baker[3,4], Sabrina Golling[2], Juliane Siebourg Polster[2], Mitchell P. Ledwith[1], Anthony Gitter[3,4,5], Angelique Augustin[2], Hassan Javanbakht[6,¤b], Andrew Mehle[1]*

1 Department of Medical Microbiology and Immunology, University of Wisconsin-Madison, Madison, Wisconsin, United States of America, 2 Roche Pharma Research and Early Development, Pharmaceutical Sciences–Biomarkers, Bioinformatics and Omics & Pathology, Roche Innovation Center Basel, F. Hoffmann-La Roche Ltd, Basel, Switzerland, 3 Department of Computer Sciences, University of Wisconsin-Madison, Madison, Wisconsin, United States of America, 4 Morgridge Institute for Research, Madison, Wisconsin, United States of America, 5 Department of Biostatistics and Medical Informatics, University of Wisconsin-Madison, Madison, Wisconsin, United States of America, 6 Roche Pharma Research and Early Development, Infectious Diseases, Roche Innovation Center Basel, F. Hoffmann-La Roche Ltd, Basel, Switzerland

¤a Current address: Lovelace Biomedical Research Institute, Albuquerque, New Mexico, United Sates of America
¤b Current address: Bluejay Therapeutics, San Francisco, California, United States of America
* sbaker@lovelacebiomedical.org (SFB); amehle@wisc.edu (AM)

**Data Availability Statement:** RNA sequencing data can found as part of BioProject PRJNA667475. Remaining data can be found in 'Supporting Information.' Quantitative data are in the

## Abstract

Viruses must balance their reliance on host cell machinery for replication while avoiding host defense. Influenza A viruses are zoonotic agents that frequently switch hosts, causing localized outbreaks with the potential for larger pandemics. The host range of influenza virus is limited by the need for successful interactions between the virus and cellular partners. Here we used immunocompetitive capture-mass spectrometry to identify cellular proteins that interact with human- and avian-style viral polymerases. We focused on the proviral activity of heterogenous nuclear ribonuclear protein U-like 1 (hnRNP UL1) and the antiviral activity of mitochondrial enoyl CoA-reductase (MECR). MECR is localized to mitochondria where it functions in mitochondrial fatty acid synthesis (mtFAS). While a small fraction of the polymerase subunit PB2 localizes to the mitochondria, PB2 did not interact with full-length MECR. By contrast, a minor splice variant produces cytoplasmic MECR (cMECR). Ectopic expression of cMECR shows that it binds the viral polymerase and suppresses viral replication by blocking assembly of viral ribonucleoprotein complexes (RNPs). MECR ablation through genome editing or drug treatment is detrimental for cell health, creating a generic block to virus replication. Using the yeast homolog Etr1 to supply the metabolic functions of MECR in MECR-null cells, we showed that specific antiviral activity is independent of mtFAS and is reconstituted by expressing cMECR. Thus, we propose a strategy where alternative splicing produces a cryptic antiviral protein that is embedded within a key metabolic enzyme.

spreadsheet S1 Data.xlsx. A separate tab is associated with each panel in the Figures and Supporting Information. Uncropped images are found in S1 Raw Images.pdf.

**Funding:** This work was supported by National Institutes of Health/National Institute for Allergy and Infectious Diseases R01AI164690 and the Greater Milwaukee Foundation Shaw Scientist Award to AM, R21AI125897 to AM and AG, a Roche Postdoctoral Fellowship RPF-353 and the National Institutes of Health/National Institute for Allergy and Infectious Diseases T32AI55397 to SFB, T32LM012413 to AB, and a National Science Foundation GRFP DGE-1747503 to MPL. AM is a Burroughs Wellcome Fund Investigator in the Pathogenesis of Infectious Disease and an H. I. Romnes Faculty Fellow funded by the Wisconsin Alumni Research Foundation. The funders had no role in study design, data collection and analysis, decision to publish, or preparation of the manuscript.

**Competing interests:** I have read the journal's policy and the authors of this manuscript have the following competing interests: AM is an editorial board member for PLoS Biology and PLoS Pathogens. HM, MT, SG, JSP, AA and HJ were employees of F. Hoffmann-La Roche when performing this work. No other authors declare a competing interest.

**Abbreviations:** ACP, acyl carrier protein; AGC, automatic gain control; AP-MS, affinity purification-MS; cMECR, cytoplasmic MECR; co-IP, co-immunoprecipitation; DDA, data-dependent acquisition; EIC, extracted ion chromatogram; HCD, high-energy collision dissociation; hnRNP UL1, heterogenous nuclear ribonuclear protein U-like 1; ICC-MS, immunocompetitive capture-mass spectrometry; IFN, interferon; IT, injection time; LC-MS/MS, liquid chromatography-tandem mass spectrometry; LTM, lactimidomycin; MAVS, mitochondrial antiviral-signaling protein; mCh, mCherry; MECR, mitochondrial enoyl CoA-reductase; mtFAS, mitochondrial fatty acid synthesis; NLuc, nanoluciferase; NP, nucleoprotein; RNP, ribonucleoprotein complex; VGM, virus growth media; WT, wild-type.

## Introduction

To move from one host to the next, viruses must overcome cross-species transmission barriers by engaging divergent cellular cofactors while evading host-encoded antiviral proteins. Emerging and reemerging influenza viruses regularly surmount host barriers through adaptive mutations that allow them to interface with new host cell environments. Migratory waterfowl are the natural host reservoir for influenza A viruses. Spillover into new hosts and adaptation has led to endemic infection in mammals including humans, pigs, dogs, and horses. Yearly, influenza virus epidemics shape public health programs worldwide. Protection through vaccination and treatment with antivirals helps slow influenza virus, yet infections still cause approximately 61,000 deaths yearly in the United States during high severity seasons [1]. Thus, it is paramount to understand what conserved cellular functions are engaged by viruses and allow them to establish infection, especially during initial cross-species infections.

Viruses are completely dependent upon cellular cofactors. The host counters this dependence and maintains pressure on the invading virus through positive selection of mutations on critical cofactors or antiviral proteins. The virus responds with its own set of adaptive mutations. The recursive process of host evasion countered by viral adaptation establishes a so-called molecular arms race, or Red Queen genetic conflict [2]. The process is aided by the fact that most host antiviral genes are nonessential for the host cell, enabling mutation without compromising viability, and mutational tolerance is further bolstered through gene duplication and transcriptional regulation [3]. Gene duplication and diversification allow hosts to mutate otherwise essential genes, whereas changes in transcriptional regulation can selectively activate genes should their constitutive expression be detrimental. To counter mutational tolerance, it has been suggested that viruses target essential genes as host cofactors, because the genes are less prone to variability [4].

Due to their rapid replication and high mutation rates relative to the host, viruses are eventually successful in both winning genetic conflicts and adapting to new hosts. For example, the human protein ANP32A or its paralog ANP32B are required for influenza viral genome replication [5]. While ANP32A/B double knockout mice are not viable, functional overlap between both proteins theoretically allows hosts to test virus escape mutations in one or the other paralog without a strong global fitness cost [6]. This is perhaps exemplified by the insertion of duplicated sequence in the avian *ANP32A* locus and a loss-of-function mutation in *ANP32B*. This has forced avian-adapted influenza polymerases to adapt and become solely reliant upon ANP32A in most avian hosts [7,8]. As a consequence, avian-adapted viral polymerases function poorly in mammals [9]. Nonetheless, a single amino acid change in the viral polymerase PB2 subunit (E627K) allows avian influenza viruses to rapidly adapt to and exploit human ANP32A [10].

The viral ribonucleoprotein complex (RNP) is the minimal unit for viral genome replication and a major hotspot for influenza virus host adaptation. RNPs are helically wound structures composed of genomic RNA encapsidated by viral nucleoprotein (NP) with the heterotrimeric RNA-dependent RNA polymerase at one end binding both the 5′ and 3′ termini of the genome. The polymerase is composed of the PB1, PB2, and PA subunits. All of the enzymatic activities required for replication and transcription are intrinsic to the polymerase, whereas host factors serve as essential cofactors or modulate RNP function [11,12]. The polymerase assumes distinct conformations during each step of the replication or transcription cycle, presenting unique interfaces for host protein interactions (reviewed in [13,14]). Indeed, MCM, ANP32A or ANP32B, and RNAP2 are host proteins that facilitate replication of the plus-sense genome intermediate cRNA, the minus-sense genomic vRNA, or transcription, respectively [5,15,16].

Understanding which host proteins the polymerase requires for replication, especially those that it engages when influenza virus jumps from one host to the next, is essential for understanding the genetic conflicts that establish barriers to cross-species transmission. Transmission of influenza virus from birds to humans requires that the incoming vRNPs successfully interact with cellular factors to, at a minimum, provide sufficient levels of replication during which adaptive mutations can arise. To understand how avian viruses engage the foreign intracellular environment of human cells, we used immunocompetitive capture-mass spectrometry (ICC-MS) to define interaction networks between human proteins and avian or human-adapted viral PB2. We identified heterogenous nuclear ribonuclear protein U-like 1 (hnRNP UL1) as a cellular cofactor that supports influenza virus replication and mitochondrial enoyl CoA-reductase (MECR) that plays an antiviral role. MECR localizes to the mitochondria and is a critical enzyme in mitochondrial fatty acid synthesis (mtFAS), raising questions as to how it could counteract the viral polymerase in the cell nucleus. Surprisingly, we demonstrated that MECR antiviral activity is derived from a cryptic splice isoform that produces cytosolic MECR (cMECR). cMECR is identical to MECR other than its lack of a mitochondrial targeting sequence. Expressing cMECR interferes with viral infection by inhibiting de novo assembly of vRNPs. Loss of MECR cripples mtFAS and cellular metabolism, resulting in defects in cell health and a generic block to viral replication. However, repairing MECR-deficient cells with the yeast homolog Etr1, which lacks a cMECR-like isoform, showed that cMECR alone suppresses influenza virus replication independent of mtFAS. Thus, we propose that *MECR* encodes a critical metabolic enzyme important for cell health while also concealing the antiviral protein cMECR that is revealed by differential splicing.

## Results

### Identification of host proteins and pathways that interface with influenza virus polymerase

During zoonoses, avian influenza virus polymerases must co-opt mammalian host processes and proteins to direct virus replication. To identify polymerase cofactors important for zoonotic and endemic transmission, we infected human lung cells with virus encoding avian-style PB2 E627 or the human style PB2 K627 and performed affinity purification-MS (AP-MS) for PB2. As these experiments were performed using infected cells, PB2 exists alone, as part of the heterotrimeric polymerase, or as part of the larger RNP. AP-MS approaches can be complicated by high levels of nonspecific interactions. We overcame these limitations by performing ICC-MS (**Fig 1A**). ICC-MS is a label-free strategy that includes a competition step with in-solution antibody prior to affinity purification using antibody already bound to a solid support. ICC-MS thus distinguishes specific interactions, which can be competed, from nonspecific interactions, which are unaffected by the competition step [17]. The competition profiles are further used to rank-order putative host interactors. Increasing amounts of competing antibody specifically reduced capture of PB2 and the viral RNP (**Fig 1B**). Avian-style PB2 E627 was more sensitive to competition, possibly because this virus is restricted in human cells and expresses lower levels of viral proteins [18]. The samples were then processed for protein identification by liquid chromatography-tandem mass spectrometry (LC–MS/MS). Effective competition was confirmed where increased competitor antibody decreased the relative abundance of PB2 itself, as well as the RNP components PB1, PA, and NP that interact with PB2 (**Figs 1C and S1A**). A total of 1,377 proteins were identified in precipitates from PB2 K627 or E627 infections at all five antibody concentrations and in at least two of three biological replicates. Hits were prioritized based on their competition curve profiles to produce a focused list of 22 candidates (**S1 Table**). We also included ANP32A, EWSR1, FUS, and GMPS

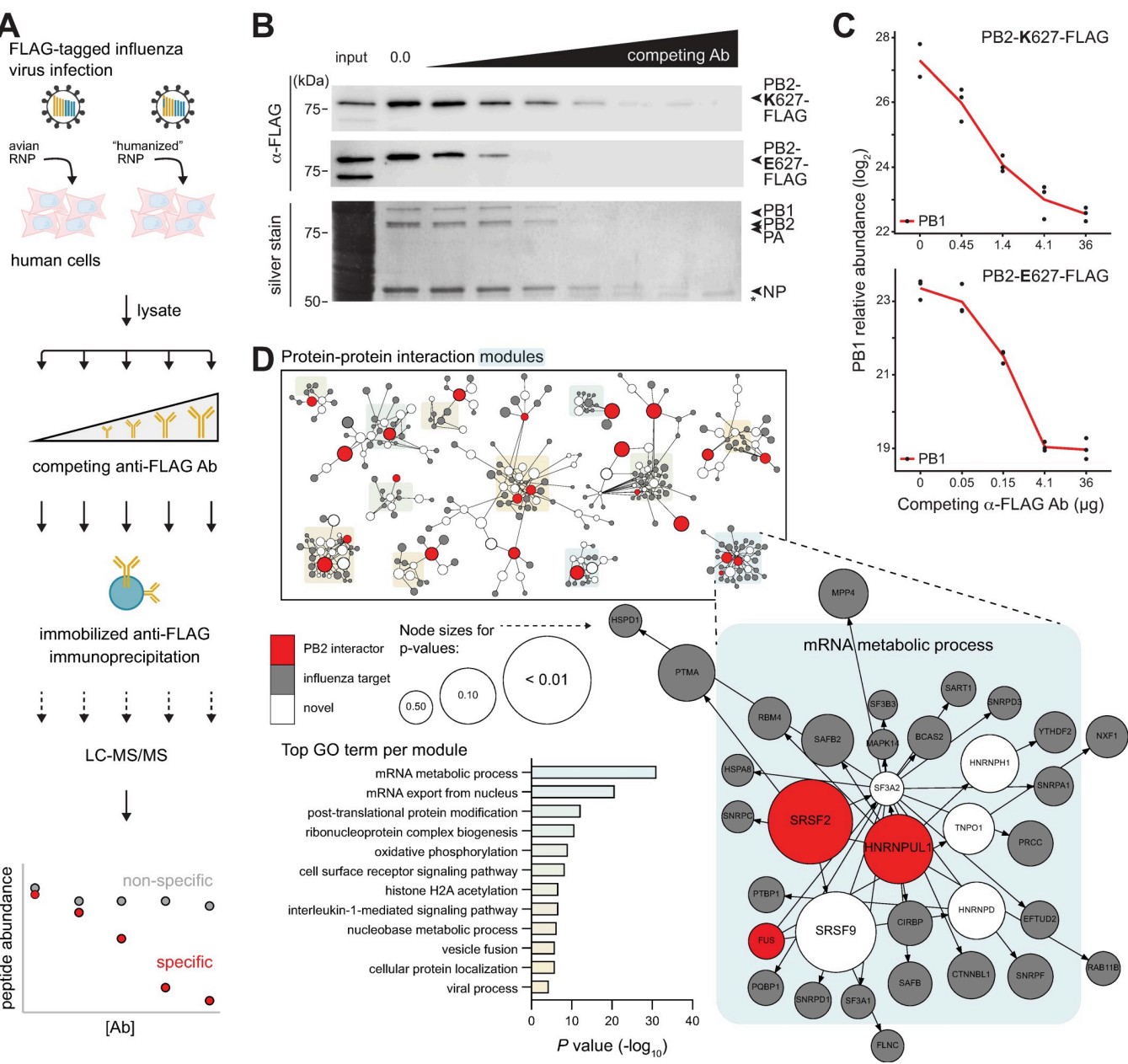

**Fig 1. ICC-MS defines an influenza polymerase interactome. (A)** Schematic diagram to identify PB2 interactome. Lysate fractions from human A549 cells infected with PB2-FLAG-tagged influenza virus encoding avian RNP or the "humanized" PB2-E627K RNP were incubated with competing soluble anti-FLAG antibody followed by capture with resin-bound anti-FLAG and LC–MS/MS. **(B-C)** Immunocompetitive capture of the viral RNP. **(B)** Detection of PB2 627E and 627K (top: western blot) or PB2 627K with coprecipitating RNP components (bottom: silver stain). *, IgG heavy chain. **(C)** Relative protein abundance of PB1 in PB2 ICC-MS samples shows decreasing capture with increasing competition antibody. Data shown are in biological triplicate. **(D)** Minimum-cost flow simulations connect top PB2 interactors identified by ICC-MS (red) to previously identified influenza host factors (gray), in some cases through other host proteins (white). Modules comprising different PB2 interactors were enriched for GO terms, the most significant of which is enlarged. Node sizes indicate empirical *P* values derived from the control flow simulations. Individual quantitative observations that underlie the data summarized here can be located under the Supporting information file as S1 Data. Uncropped images can be found in the Supporting information file as S1 Raw Images. Ab, antibody; ICC-MS, immunocompetitive capture-mass spectrometry; IgG, immunoglobulin G; LC-MS/MS, liquid chromatography-tandem mass spectrometry; NP, nucleoprotein; RNP, ribonucleoprotein complex.

that were identified in a pilot screen, but not among the top candidates in the subsequent analysis. Several of the PB2 ICC-MS interactors, including ADAR, ANP32A, ATP7A, and KPNA3, were previously identified in proteomic screens for influenza virus polymerase cofactors and studied in detail, providing confidence in our approach [5,19–21]. The majority of PB2 interactors were found during infection with avian- or human-style polymerases, suggesting the identification of cofactors conserved for both zoonotic and endemic infections.

To increase the power of our ICC-MS results, we performed protein–protein interaction network analyses tailored for influenza virus. Networks were constructed of experimentally demonstrated protein–protein interactions (STRING; [22]). We then performed minimum-cost flow simulations where the candidate PB2 interactors served as sources to link to targets marked as influenza host factors based on data from six genome-wide screens [23–29]. Twenty-three of the 26 PB2 interactors identified by ICC-MS [23–29] readily formed subnetworks with previously identified influenza host factors and revealed key cellular processes defined by GO enrichment scores for each of the 12 subnetworks (**Figs 1D and S1B and S2A and S2B Table**). Further, they uncovered new protein partners within these subnetworks and connected to highly significant modules not previously associated with influenza virus.

Flow simulations can be biased due to nodes forming spurious links in order to reach large multipartner targets. But, as detailed in the Methods, the PB2 subnetworks were specific to simulations programmed with PB2 interactors and based on influenza virus host factors (**S2 Table**). To further test the networks, we queried them using host proteins with thoroughly studied interactions as sources: PKR and RIG-I, proteins important for innate immune defense against influenza virus (reviewed in [30,31]); CRM1 and NXF1, proteins important for viral nuclear-cytoplasmic transport [32,33]; and EXOSC3 and UBR4, proteins that were identified through rigorous omics approaches [4,34]. These six proteins were used as sources to connect to influenza host factor targets, then controlled through two separate simulations as above. Both flow simulations produced similar results that recapitulate in silico the biochemically defined interaction networks and suggest new proteins that may be important for these processes (**S1D Fig and S2C and S2D Table**).

## *HNRNPUL1* promotes and *MECR* restricts viral infection

To test the functional role of proteins identified by ICC-MS, candidate interactors were knocked down by siRNA treatment in A549 cells prior to infection with influenza virus (**Fig 2A**). Infections were performed with human-style PB2 K627 and avian-style E627 viruses to detect any species-specific dependence. NXF1, an essential host cofactor, was knocked down as a positive control and caused a severe decrease in replication, as expected [24]. Knockdown of hnRNP UL1 reduced viral titers to approximately 25% of the nontargeting control, whereas knockdown of MECR significantly increased titers 2.5- to 5-fold. Knockdown of other interactors had only modest effects in A549 cells, possibly because these factors function redundantly (e.g., importin-α isoforms, ANP32A and ANP32B), are needed in very limited quantities (e.g., ANP32A), or are not essential for viral replication [20,35]. Titers in the knockdown cells for virus encoding PB2 K627 or PB2 E627 were highly correlated (Pearson's $r^2 = 0.791$), suggesting that our interactors had comparable roles for human-signature or avian-signature virus polymerases (**Fig 2B**).

Separate experiments confirmed knockdown of hnRNP UL1 and MECR (**Fig 2C**). Reductions in hnRNP UL1 protein levels decreased viral replication, whereas reduction in MECR protein levels increased titers. Similar knockdown phenotypes were detected during multicycle replication in another human cell line, 293T (**S2A and S2B Fig**). We extended these findings to primary isolates of influenza A virus from the 2009 pandemic (A/California/04/2009

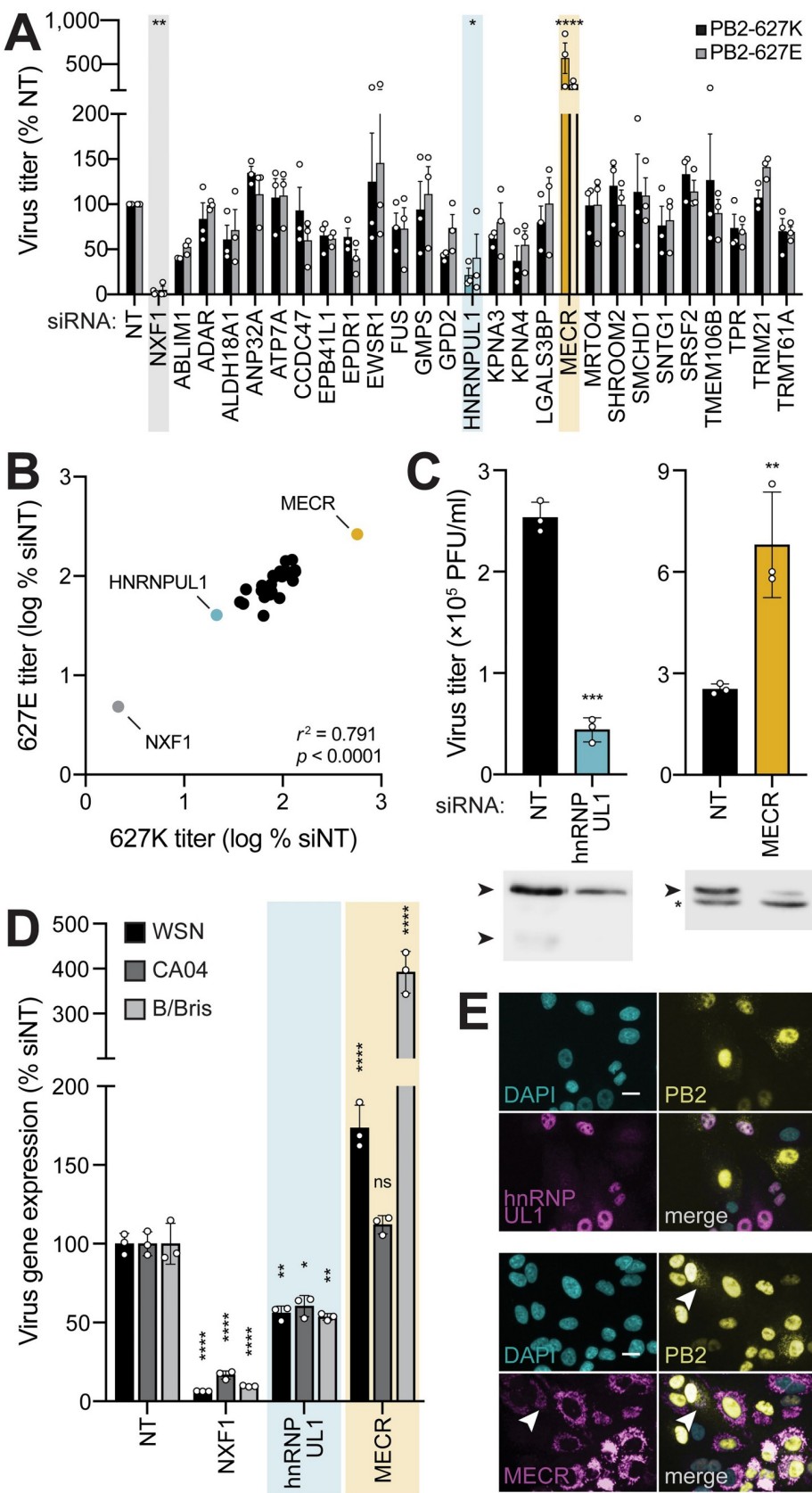

**Fig 2. Functional analysis of top candidate PB2 interactors reveals important roles for hnRNP UL1 and MECR.**
**(A)** Secondary screening of proteomic hits by siRNA treatment and reporter virus infection. After knockdown, A549 cells were infected with human (PB2-627K; MOI, 0.01) or avian-adapted (PB2-627E; MOI, 0.05) WSN NLuc virus for 24 h. Viral supernatants were titered and normalized to an NT control. Control NXF1 (gray) and outliers hnRNP UL1 (cyan) and MECR (yellow) are highlighted. **(B)** Concordance of virus titer for PB2-627E vs. PB2-627K virus infections in siRNA-treated cells (from **(A)**). Statistical analysis performed with a two-tailed Pearson correlation coefficient. **(C)** Multicycle virus replication of WT virus was measured 24 h postinfection in A549 cells treated with the indicated siRNAs. Knockdown efficiency was analyzed by western blot. Asterisk indicates nonspecific band. **(D)** Knockdown impacts viral gene expression of divergent influenza viruses. siRNA-treated A549 cells were infected with reporter viruses based on WSN (pre-2009 H1N1; MOI, 0.1), CA04 (pandemic 2009 H1N1; MOI, 0.5), or B/Bris (Victoria-lineage; MOI, 1). Viral gene expression was measured 8 h postinfection and normalized to NT controls. **(E)** A549 cells stably expressing hnRNP UL1 (top) or MECR (bottom) were infected with WSN PB2-FLAG (MOI 3, 8 h). Protein localization was detected by immunofluorescence, and nuclei were visualized with DAPI. Arrow indicates minor PB2 population consistent with previously reported patterns of mitochondrial localization. Scale bars, 20 μm. Data in **(A)** are mean ± SEM of $n = 3$ biological replicates. Comparisons were performed with two-way ANOVA with post hoc Fisher LSD test. For **(C)** and **(D)**, data are mean ± SD of $n = 3$. Comparisons were performed with a two-tailed Student $t$ test **(C)** or a two-way ANOVA with post hoc Dunnett multiple comparisons test **(D)**; *, $P < 0.05$; **, $P < 0.01$; ***, $P < 0.001$; ****, $P < 0.0001$; ns, not significant. Individual quantitative observations that underlie the data summarized here can be located under the Supporting information file as S1 Data. Uncropped images can be found in the Supporting information file as S1 Raw Images. hnRNP UL1, heterogenous nuclear ribonuclear protein U-like 1; MECR, mitochondrial enoyl CoA-reductase; NLuc, nanoluciferase; NT, nontargeting; WT, wild-type.

[CA04]; H1N1) and influenza B virus (B/Brisbane/60/2008 [B/Bris]) (**Fig 2D**). Loss of hnRNP UL1 reduced viral gene expression during infection for all strains, as did our control target NXF1. Knockdown of MECR again increased infection by WSN, but not CA04. B/Bris was even more impacted by MECR knockdown with an approximately 4-fold increase in viral gene expression compared to 1.5- to 2-fold effect seen for WSN. We also measured viral gene expression during a single round of infection. Similar to results with viral replication, viral gene expression was reduced when hnRNP UL1 was knocked down and increased when MECR was knocked down, and this was independent of the identity of PB2 residue 627 (**S2C and S2D Fig**). These data suggest that hnRNP UL1 functions as a proviral factor, contrasting with the antiviral activity associated with MECR expression.

## hnRNP UL1 interacts with the viral replication machinery to promote replication

hnRNP UL1 is an RNA-binding protein that plays a role in nucleocytoplasmic RNA transport as well as DNA end resection signaling during double strand break repair [36,37]. As a member of the hnRNP family of proteins that are well-characterized regulators of pre-mRNA processing, hnRNP UL1 has been shown to interact with NXF1 and NS1-BP, proteins that help coordinate export of influenza viral mRNAs and splicing of the viral genome, respectively [33,38,39]. Consistent with its known function, immunofluorescence assays showed that hnRNP UL1 is present primarily in the nucleus of infected cells where it colocalized with PB2 (**Fig 2E**). Infection does not appear to change hnRNP UL1 localization. We tested interactions between the viral polymerase and endogenous hnRNP UL1. Cells were infected with wild-type (WT) virus or virus encoding PB2-FLAG and subject to FLAG immunoprecipitation. Endogenous hnRNP UL1 coprecipitated in the presence of PB2-FLAG, as did the polymerase subunit PB1, but neither were present in the control precipitation with untagged PB2 (**Fig 3A**), demonstrating specific association with PB2 and confirming our ICC-MS.

Nucleocytoplasmic transport of viral mRNA is a major bottleneck for viral gene expression. There is a limiting amount of host NXF1, which chaperones mature viral mRNA to the nuclear basket for transport [33,39]. hnRNP UL1 forms RNP complexes with NXF1 [39]. To test the importance of hnRNP UL1 and whether it is limiting, we stably overexpressed hnRNP UL1. Viral titers increased almost 3-fold in cells expressing more hnRNP UL1 (**Fig 3B**). Similar

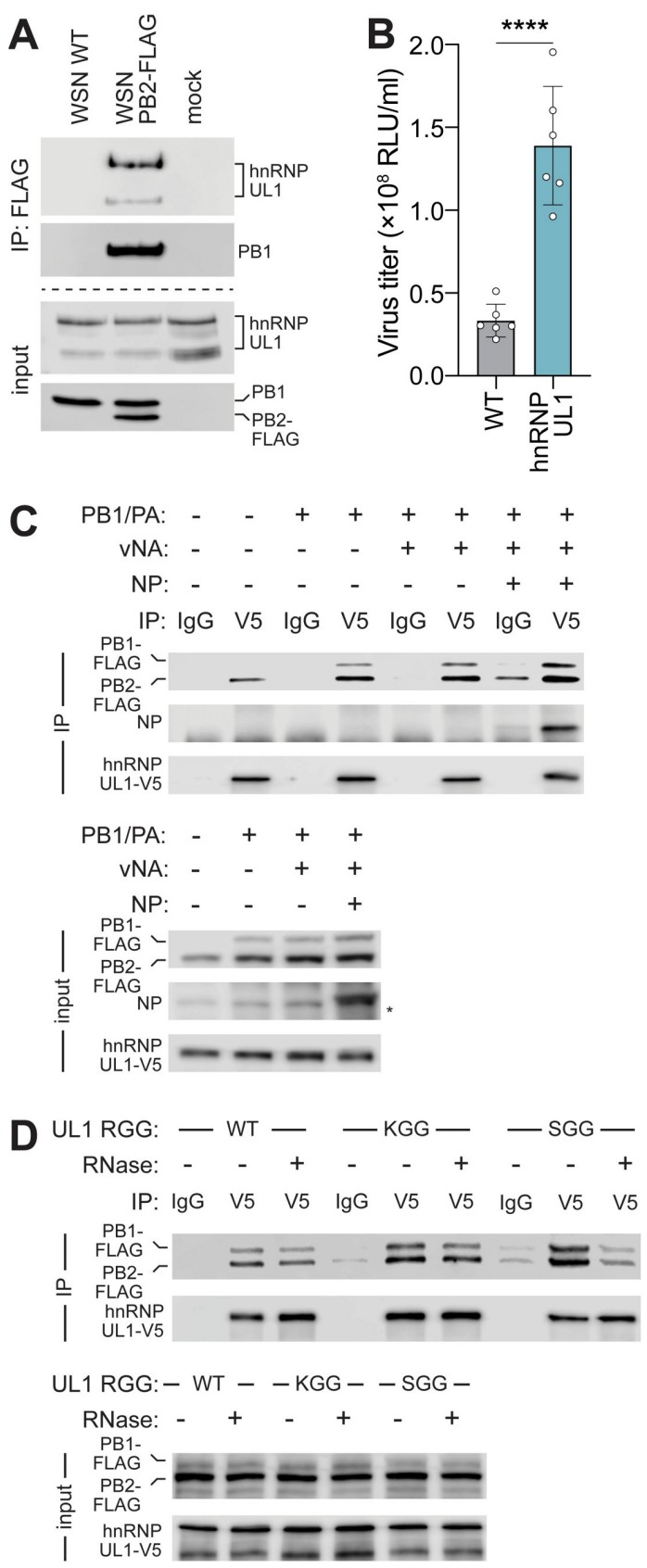

**Fig 3. Proviral hnRNP UL1 associates with influenza polymerase. (A)** Endogenous hnRNP UL1 coprecipitates with PB2 during infection. A549 cells were infected (MOI, 1; 18 h) or mock treated, lysed, and immunoprecipitated. Proteins were detected by western blot. **(B)** Viral titers were measured from WT or clonal A549 hnRNP UL1-V5 cells infected with WSN Nluc (MOI, 0.05; 24 h). Mean ± SD of $n = 6$. Unpaired two-tailed $t$ test; ****, $P < 0.0001$. **(C-D)** Association of viral polymerase with hnRNP UL1. **(C)** hnRNP UL1-V5 was immunoprecipitated with anti-V5 antibody or control IgG from lysates generated from cells expressing the indicated viral proteins or RNA (vNA). Proteins were detected by western blotting. Asterisk indicates nonspecific band. **(D)** Co-immunoprecipitation assays were performed using WT hnRNP UL1 or mutants disrupted in methylation (KGG) or RNA binding (SGG). RNase was included during immunoprecipitation where indicated. Individual quantitative observations that underlie the data summarized here can be located under the Supporting information file as S1 Data. Uncropped images can be found in the Supporting information file as S1 Raw Images. hnRNP UL1, heterogenous nuclear ribonuclear protein U-like 1; IgG, immunoglobulin G; IP, immunoprecipitation; NLuc, nanoluciferase; NP, nucleoprotein; WT, wild-type.

results were demonstrated upon overexpression of NXF1 or TPR, which connects viral mRNA to the nuclear pore complex, further demonstrating that nuclear export is a limiting step during infection (S3A Fig; [40]). The viral polymerase is an RNA-binding protein that exists both in a free form or incorporated into viral RNP with genomic RNA and NP. We performed a series of interactions studies to dissect the complexes that interact with hnRNP UL1 and the role of RNA in these interactions. hnRNP UL1 interacted with PB2 when PB2 was expressed by itself (**Fig 3C**). hnRNP UL1 showed more robust interactions when all three polymerase subunits were present, indicating that hnRNP UL1 interacts with the trimeric polymerase in the absence of other viral proteins or RNAs. Expressing genomic RNA in these cells did not increase the interaction between the polymerase and hnRNP UL1. However, including genomic RNA and NP, which permits formation of viral RNP, resulted in specific coprecipitation of NP (**Fig 3C**). hnRNP UL1 is also an RNA-binding protein that forms multiprotein complexes. hnRNP UL1 binds RNA through its RGG box and methylation of arginine residues in the RGG box facilitates association with some of its protein partners [41,42]. To evaluate these functions, we mutated hnRNP UL1 to eliminate RNA binding (RGG to SGG) or methylation of the RGG box (RGG to KGG) (**Fig 3D**). Interactions between the viral polymerase and hnRNP UL1 were indistinguishable between WT, an RNA-binding mutant, or a methylation mutant. These mutations did not change the normal nuclear localization of hnRNP UL1 (**S3B Fig**). To further test for a role of bridging RNA, exogenous RNase was added during immunoprecipitation. RNase treatment resulted in minor changes for WT and the RGG, while complex formation appeared reduced for the SGG mutant, suggesting that this variant may rely more on RNA bridging to engage the polymerase (**Fig 3D**). Together, these results show that hnRNP UL1 enhances replication by binding to the viral polymerase and replication complex, potentially increasing access to rate-limiting pathways.

## *MECR* antiviral activity is independent from its role in mtFAS

MECR is an essential enzyme for mtFAS. The acyl carrier protein (ACP) encoded by *NDU-FAB1* scaffolds each enzymatic step of mtFAS in the mitochondria with MECR performing the final step converting *trans*-2-enoyl-ACP to acyl-ACP (**Fig 4A**; reviewed in [43]). Acyl-ACP feeds into oxidative phosphorylation or is converted to octanoyl-ACP then lipoic acid for protein lipoylation or use in the TCA cycle. To test whether the mtFAS pathway itself has an antiviral role, we knocked down ACP, which potently limits lipoic acid accumulation [44]. Contrary to the antiviral activity of MECR, knockdown of ACP did not cause a significant change in viral titers (**Fig 4B**). Combined knockdown of MECR and ACP maintained the increased replication associated with MECR knockdown but did not have any additive effects. We additionally targeted mtFAS by treating infected cells with C75, a drug that targets OXSM in the mtFAS pathway as well as FASN that directs cytoplasmic fatty acid synthesis [45].

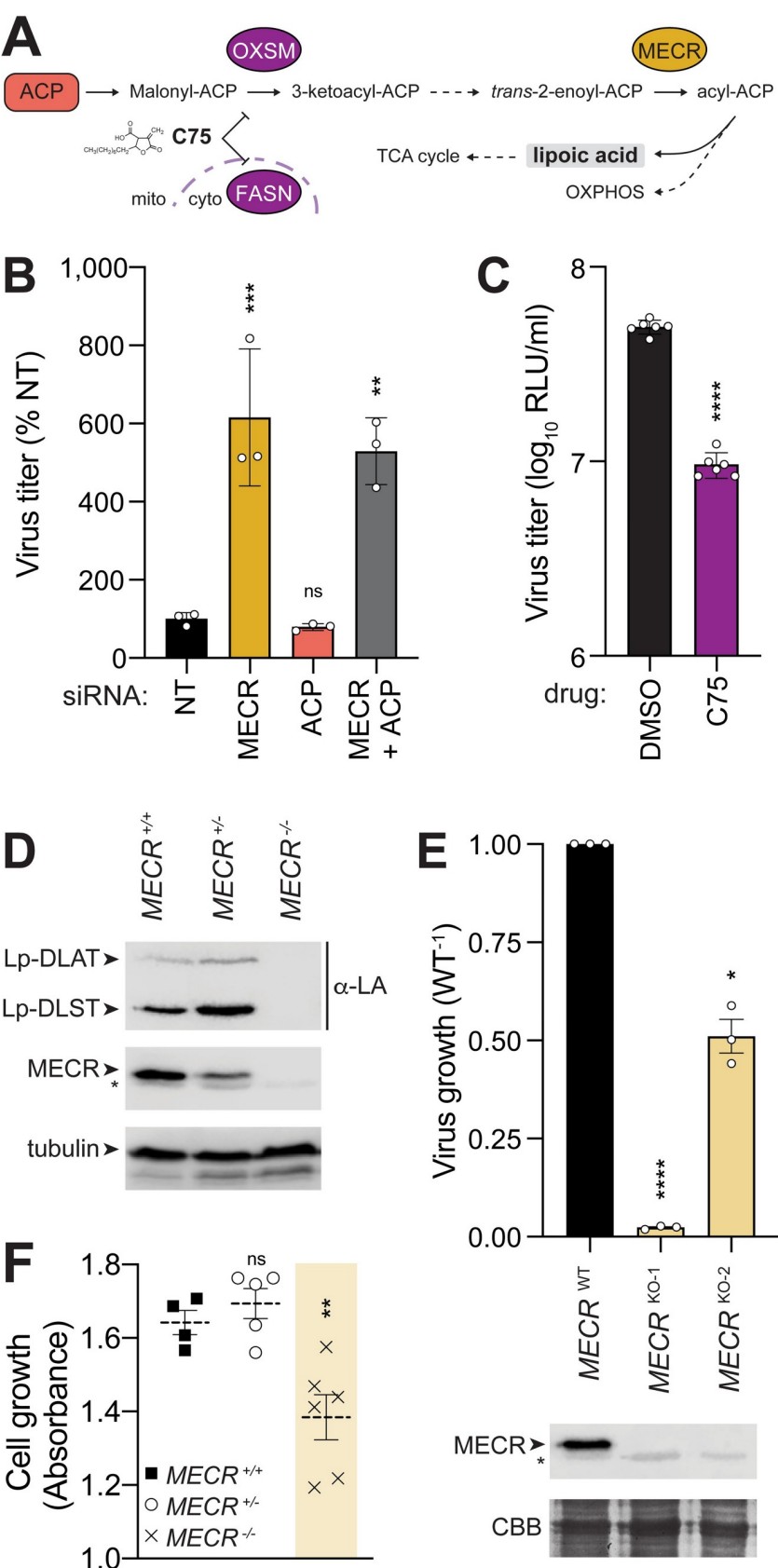

**Fig 4. Modulating the critical mtFAS enzyme MECR alters mtFAS output and virus growth. (A)** Focused snapshot of the mtFAS pathway. ACP and MECR were experimentally probed by knockdown or knockout, whereas OXSM and FASN were inhibited with the drug C75. **(B)** A549 cells were treated with siRNA targeting MECR, ACP, both, or an NT control prior to infection with WSN NLuc virus (MOI, 0.05; 24 h). Viral titer in supernatants was determined and normalized to NT. Mean ± SD of $n = 3$. One-way ANOVA with post hoc Dunnett multiple comparisons test; **, $P < 0.01$; ***, $P < 0.001$; ns, not significant. **(C)** Viral yield was measured from A549 cells treated with C75 or DMSO control prior to infection with WSN NLuc (MOI, 0.05; 24 h). Mean ± SD of $n = 6$. Unpaired two-tailed $t$ test; ****, $P < 0.0001$. **(D)** Production of lipoylated subunits of the pyruvate dehydrogenase complex (DLAT) and the 2-oxoglutarate dehydrogenase complex (DLST) was assessed in WT ($^{+/+}$), heterozygous ($^{+/-}$), and homozygous ($^{-/-}$) *MECR* knockout A549 cells by western blotting with anti-lipoic acid ($\alpha$-LA) antibody. **(E)** Virus replication was measured in A549 cells (MOI, 0.05; 24 h). Replication in MECR knockout clones KO-1 and KO-2 was normalized to WT A549 cells. Mean ± SEM of biological replicates ($n = 3$) normalized to WT. MECR expression was monitored by western blotting. CBB staining was used as a loading control. Asterisks indicate nonspecific bands. **(F)** Growth of clonal WT, heterozygous, or homozygous *MECR* knockout A549 cells was measured over 3 d. Mean ± SEM of $n = 4$–6 clones. One-way ANOVA with post hoc Dunnett multiple comparisons test; **, $P < 0.01$; ns, not significant. Individual quantitative observations that underlie the data summarized here can be located under the Supporting information file as S1 Data. Uncropped images can be found in the Supporting information file as S1 Raw Images. ACP, acyl carrier protein; CBB, Coomassie brilliant blue; DLAT, dihydrolipoamide acetyltransferase; DLST, dihydrolipoamide S-succinyltransferase; FASN, fatty acid synthase; MECR, mitochondrial enoyl CoA-reductase; mtFAS, mitochondrial fatty acid synthesis; NLuc, nanoluciferase; NT, nontargeting; OXSM, 3-oxoacyl-ACP synthase, mitochondrial; WT, wild-type.

Whereas specific knockdown of MECR increased replication, complete loss of fatty acid synthesis by C75 treatment reduced viral titers, likely by disrupting cellular metabolism (Fig 4C).

To further dissect the antiviral role of MECR and any contribution from mtFAS, we generated MECR knockout A549 cells. *MECR* is not a strictly essential gene in cell culture [46], but deletions in mice are embryonic lethal and it may be necessary for mammalian skeletal myoblasts [47,48]. We recovered A549 cells with edited heterozygotic and homozygotic knockout alleles (S4A Fig). Loss of MECR completely disrupted mtFAS, as indicated by loss of lipoylated proteins (Fig 4D). mtFAS remained intact in a heterozygotic cell clone. Our knockdown experiments predicted that loss of MECR would enhance virus production, yet we observed that two independent MECR knockout cells uniformly produced less virus than WT (Fig 4E). *MECR* knockout cells grew slower compared to WT or *MECR* heterozygotic cells (Fig 4F). The limited virus growth in knockout cells is likely not connected to the antiviral activity of MECR but is instead due to cellular defects in mtFAS that manifest as a loss of mitochondrial lipoic acid synthesis, slower cell growth, and defects in respiratory chain complex integrity (Fig 4E and 4F; [44]). These data suggest that MECR antiviral activity is independent from its normal role in supporting cellular metabolism.

## A cryptic isoform of *MECR* localizes to the cytoplasm and targets the polymerase to disrupt viral RNP assembly

Our ICC-MS results provided high confidence data demonstrating interactions between endogenous MECR and PB2 during viral infection (S1 Table). While a minor population of PB2 localizes to the mitochondria (Fig 2D; [49]), we did not detect robust interactions between the polymerase and full-length MECR (S5D Fig). Analysis of RNA-seq data from influenza virus infected A549 cells revealed that 29% of MECR transcripts (± 5%, $n = 3$) have an alternative splicing pattern. The alternative transcript form utilizes an upstream splice acceptor site between exons 1 and 2 (Fig 5A), which encodes multiple upstream open reading frames and stop codons in all frames, thus creating an extensive untranslated region that may promote initiation at the downstream start site M77 in MECR. Reverse transcription PCR (RT-PCR) demonstrated the presence of long and short isoforms of MECR transcripts in mock and infected cells (S5A and S5B Fig).

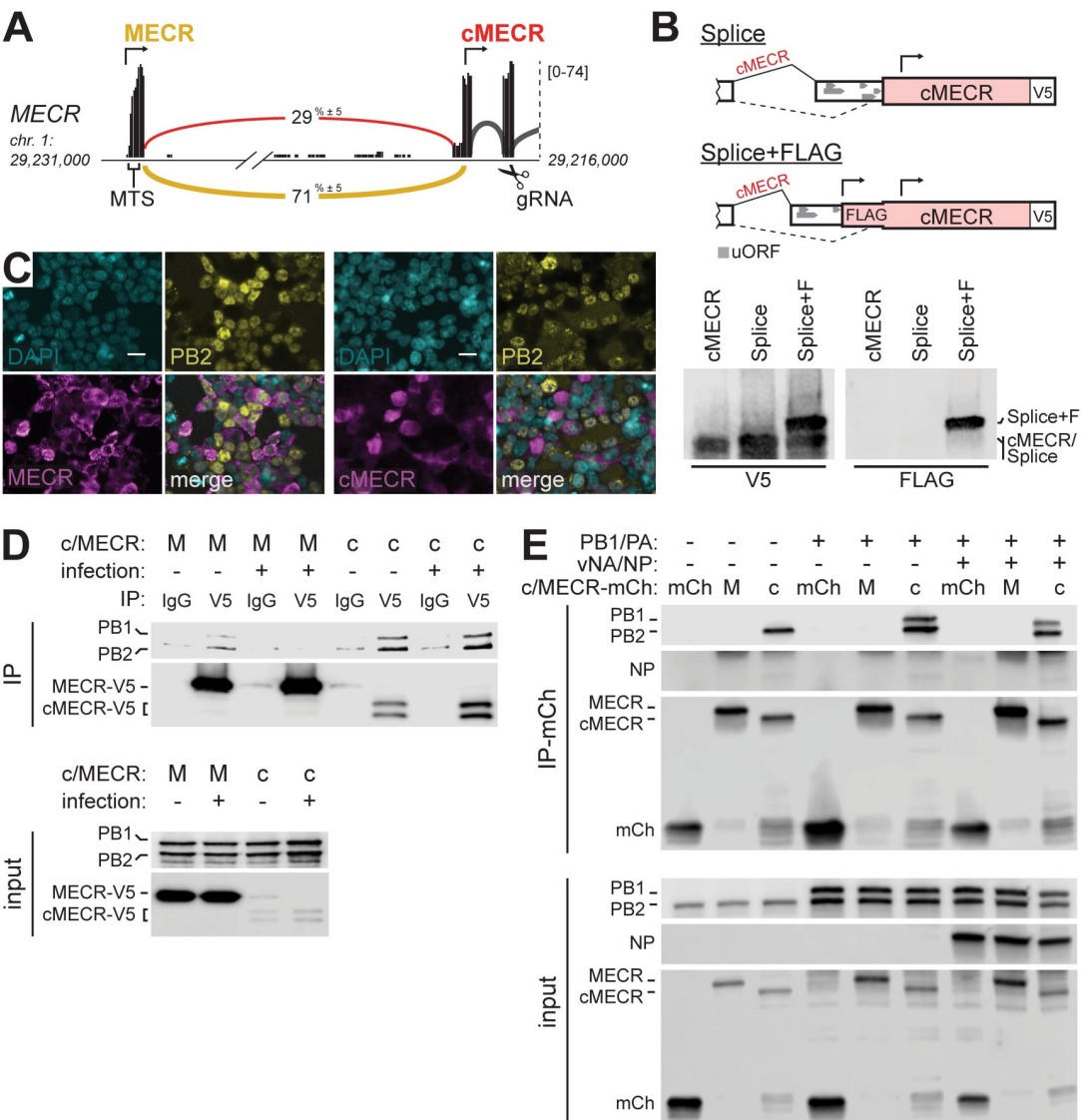

**Fig 5. The alternative splice variant cMECR binds viral polymerase subunit PB2. (A)** Sashimi plot of RNA-seq data from A549 cells showing alternative 3′ splice site utilized by transcripts that do not translate the MTS in exon 1. Numbers embedded in yellow (MECR) and red (cMECR) curves indicate percentage of each splicing event as a total of all exon-joining reads. Arrows indicate translational start sites and gRNA denotes region targeted in CRISPR-Cas9 editing. **(B)** Splicing reporters were constructed as diagrammed and expressed in 293T cells with cMECR cDNA as a control. Proteins were detected by western blot. **(C)** 293T cells expressing MECR or cMECR were infected with influenza PB2-FLAG virus (MOI, 3; 8 h). Subcellular localization of PB2, MECR, or cMECR were determined by immunofluorescence microscopy. Scale bars, 20 μm. **(D-E)** The viral polymerase subunit PB2 associates with cMECR during infection. **(D)** PA, PB1-FLAG, PB2-FLAG, and V5-tagged MECR or cMECR were expressed in 293T cells. Where indicated, cells were also infected with WSN (MOI, 10; 6 h). Cells were lysed and immunoprecipitated with anti-V5 antibody or IgG controls. Proteins were detected by western blot. **(E)** mCh, MECR M77L-mCh, or cMECR-mCh were coexpressed with PB2-FLAG in the absence or presence of PB1-FLAG, PA, NP, and vNA. Cells were lysed and immunoprecipitated with mCh affinity resin. Proteins were detected by western blot. Uncropped images can be found in the Supporting information file as S1 Raw Images. cMECR, cytoplasmic MECR; IgG, immunoglobulin G; IP, immunoprecipitation; mCh, mCherry; MECR, mitochondrial enoyl CoA-reductase; MTS, mitochondrial targeting signal; NP, nucleoprotein; vNA, *NA* viral RNA.

Translation from the in-frame downstream start codon would skip coding sequences for the mitochondrial targeting signal in MECR to produce cytoplasmic MECR, previously identified as cMECR [50]. Analysis of ribosome profiling data demonstrated translation initiation

for cMECR at downstream sites on the alternatively spliced transcript (**S5C Fig**) [51]. Initiating ribosomes were enriched in these experiments by treating cells with lactimidomycin (LTM), a translation inhibitor that arrests ribosomes at initiation sites by blocking the first translocation cycle [52]. Under these conditions, initiating ribosomes were detected at the cMECR start site and other start sites in the UTR of the alternatively spliced transcript (**S5C Fig**). These data suggest translation of cMECR. However, cMECR is a minor variant and differs from MECR only by a small change in molecular weight, preventing us from unambiguously detecting endogenous cMECR by western blot. We therefore created splicing reporters to experimentally verify translation of cMECR. The 3′ end of exon 1 and a miniaturized version of intron 1 from the *MECR* locus were cloned upstream of the minor splice acceptor site and the remainder of the MECR/cMECR cDNA (**Fig 5B**). The reporter is designed to require splicing for cMECR production as multiple stop codons in all three frames in the intron would prevent initiation and readthrough on unspliced transcripts. The splicing reporter expressed protein similar to that from cMECR cDNA (**Fig 5B**). To demonstrate that initiation occurs at the downstream start sites despite the presence of uORFs, we created another reporter that introduces an in-frame FLAG tag only if initiation occurs in the cMECR-specific UTR (**Fig 5B**). This reporter also produced cMECR, which is slightly larger due to the appended FLAG tag as shown by blotting. These data provide multiple lines of evidence suggesting that MECR transcripts are alternatively spliced and initiate translation at a downstream site to produce cMECR.

Full-length MECR displayed a mitochondrial-like subcellular organization, whereas expressing cMECR resulted in diffuse staining throughout the cytoplasm and nucleus, consistent with the absence of a mitochondrial targeting sequence (**Fig 5C**). Comparing infected and uninfected cells indicated that infection did not impact MECR or cMECR distribution. Given that cMECR cannot localize to the mitochondria, it likely does not have a role in mtFAS.

As the ICC-MS results could not distinguish whether PB2 was bound by MECR or cMECR, we individually tested these interactions by co-immunoprecipitation (co-IP). MECR was strongly expressed but coprecipitated very small amounts of PB2 when PB2 was expressed alone (**S5D Fig**). cMECR, by contrast, was poorly expressed but robustly interacted with PB2 in the absence or presence of viral polymerase and RNP components. To evaluate if infection played a role in this interaction, parallel immunoprecipitations were performed in cells coexpressing RNP and MECR or cMECR that were additionally infected with influenza virus (**Fig 5D**). Infection did not change the interactions; cMECR, but not MECR, interacted with the viral polymerase.

The minor copurification of polymerase with MECR may be due to leaky scanning and initiation from M77 in MECR, recreating cMECR from the MECR transcript, which can be seen as faint lower molecular weight bands in immunoprecipitations (**Figs 5D and S5D**). We introduced an M77L mutation in *MECR* to overcome leaky initiation. We also codon-optimized constructs to improve cMECR expression and fused them to mCherry (mCh) for visualization. These changes did not alter the subcellular localization of the expressed proteins (**S5E Fig**). Immunoprecipitations were repeated with these new constructs that strictly express MECR or cMECR. cMECR, but not MECR, again coprecipitated PB2 when it was expressed alone and within the context of the trimeric viral polymerase (**Fig 5E**). cMECR also coprecipitated PB1 when it was present, although NP was notably absent from these interactions. We utilized proximity ligation as an orthogonal approach to query interactions with cMECR. Polymerase was coexpressed with cMECR fused to the biotin ligase AirID [53]. In the presence of cMECR-AirID, purified polymerase was heavily biotinylated on PB2 and PA and, to a lesser extent, on PB1 (**S5F Fig**). These targeted interaction studies, combined with endogenous interactions detected by ICC-MS, provide multiple lines of evidence that PB2 interacts exclusively with cMECR.

Given that cMECR interacted with the viral polymerase, we asked whether it conferred the antiviral phenotype revealed by our knockdown experiments (Fig 2A and 2C). Viral titers were measured following infection of clonal A549 cells expressing MECR or cMECR. Overexpression of MECR did not alter virus replication, whereas exogenous cMECR resulted in a nearly 10-fold reduction in viral titers (Fig 6A). Successful infection requires de novo formation of viral RNPs to amplify transcription and replicate the viral genome. cMECR coprecipitated polymerase in cells where RNPs are formed, yet NP was not coprecipitated, suggesting that cMECR interacted with free polymerase but not polymerase assembled into RNPs (Fig 5E). We thus tested whether cMECR interferes with RNP assembly (Fig 6B). Lysates were prepared from cells where RNPs were assembled in the presence of MECR M77L, cMECR, or the control mCh. The lysates were split and fractions were used to assess interactions with cMECR or RNPs. In one fraction, cMECR again coprecipitated the viral polymerase, but not NP. MECR and the mCh control did not interact with viral proteins, confirming specificity. In the other fraction, NP coprecipitated the polymerase, confirming RNP formation. However, expressing cMECR reduced RNP formation compared to MECR or the mCh control. Moreover, the NP-captured RNP complexes were devoid of cMECR. To further test the ability of cMECR to disrupt RNP formation, reciprocal immunoprecipitations were performed where the viral polymerase was captured and probed for coprecipitating NP (Fig 6C). cMECR again reduced RNP formation as seen by lower amounts of NP interacting with the polymerase compared to MECR. These complementary analyses suggest mutually exclusive protein complexes where cMECR interacts with free polymerase and prevents its incorporation into RNPs.

If cMECR interferes with RNP assembly, it should also decrease polymerase activity. RNPs assembled in the above experiment contained a model vRNA encoding GFP. Polymerase activity was decreased in cells expressing cMECR, as seen by a reduction in GFP protein levels compared to cells expressing MECR M77L or the control mCh (Fig 6B, input). Coexpressing cMECR also caused an approximately 50% decrease in GFP-positive cells compared to the mCh control, while full-length MECR M77L had no appreciable effect (Figs 6D and S6A). Collectively, these data suggest that cMECR exerts its antiviral activity by suppressing RNP assembly and downstream polymerase activity.

## Repairing mtFAS with the yeast homolog assigns antiviral activity to cMECR

MECR plays an essential role in mtFAS, making it challenging to cleanly delineate MECR function during mtFAS from the antiviral role of cMECR. Indeed, mtFAS deficiency in *MECR*⁻/⁻ cells resulted in a generic defect in cellular metabolism that reduced cell growth and, consequently, viral replication (Fig 4). We therefore sought to repair the mtFAS pathway in *MECR*⁻/⁻ cells by expressing a homolog of MECR that lacks cMECR. cMECR translation relies on the start codon at M77, which is surprisingly well conserved; phylogenetic analysis suggests that the M77 start codon driving cMECR expression was acquired after the divergence of animalia from fungi (Fig 6E). Only the ancestral homolog of MECR from baker's yeast (*Saccharomyces cerevisiae*) and salmon (*Salmo salar*) and trout (*Salmo trutta*) do not encode M77 and presumably do not produce cMECR. cMECR expression also requires an alternatively spliced transcript. Annotated transcripts that code for cMECR were identified across many influenza virus hosts including pigs, mice, and geese, providing further evidence for cMECR expression (Fig 6E and S4 Table).

To assess whether the antiviral activity of cMECR is independent from mtFAS and MECR in general, we complemented *MECR*⁻/⁻ cells by stably expressing the yeast homolog Etr1. As before, knockout of MECR disrupted mtFAS as shown by the loss of lipoylated DLST and

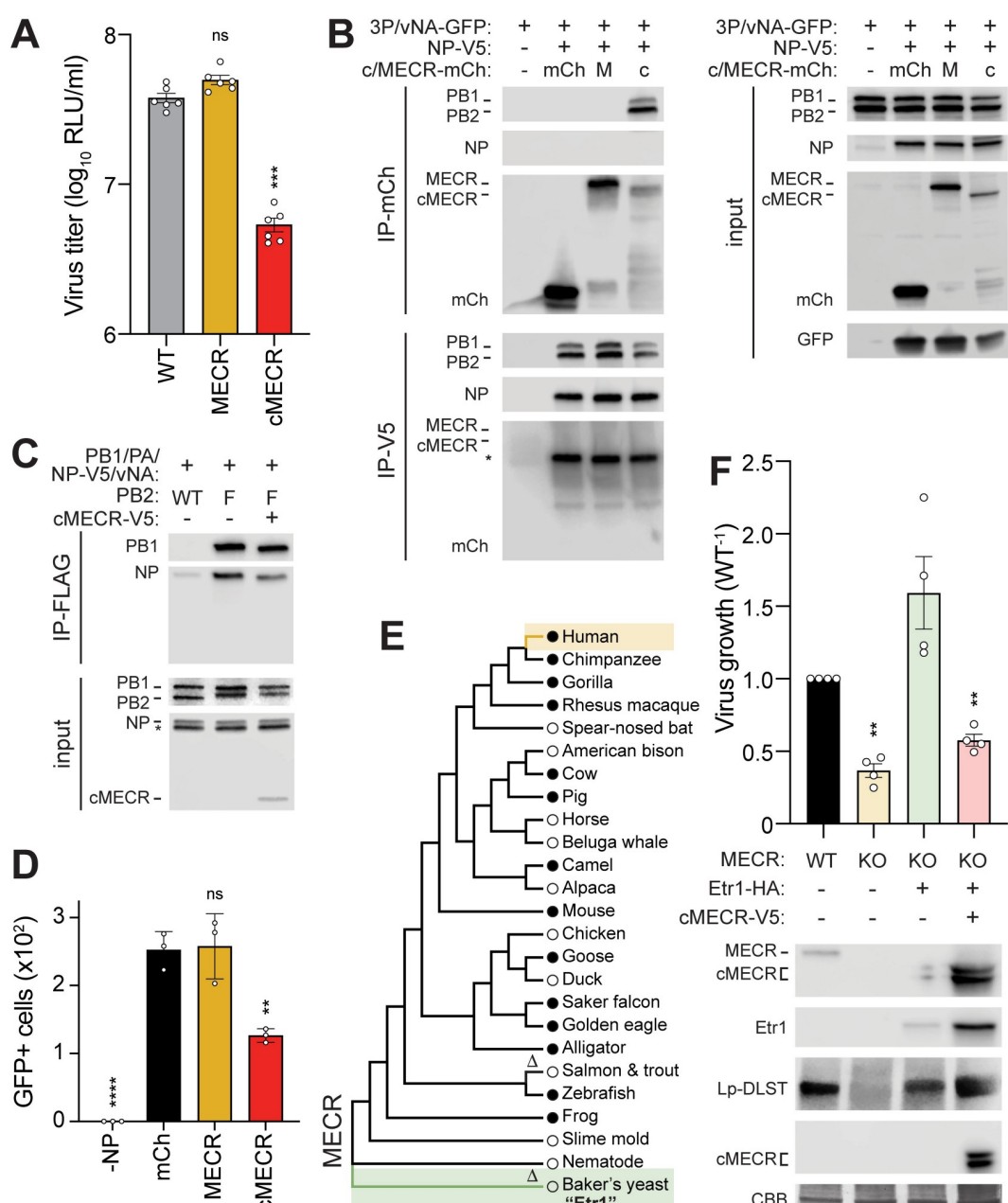

**Fig 6. The antiviral activity of cMECR inhibits RNP assembly independent of MECR and mtFAS. (A)** Replication of WSN NLuc was measured in WT or clonal A549 cells expressing MECR or cMECR. Protein expression from infected cells was analyzed by western blotting. Asterisk indicates a nonspecific band used as a loading control. Mean ± SD of $n = 6$. One-way ANOVA with post hoc Dunnett multiple comparisons test; ***, $P < 0.001$; ns, not significant. **(B-C)** cMECR disrupts RNP assembly. **(B)** RNPs were assembled in 293T cells coexpressing mCh, MECR M77L-mCh, or cMECR-mCh. NP was omitted in the negative control (−NP). Cells were lysed, divided in half, and immunoprecipitated for MECR (mCh) or NP (V5). Input or coprecipitated proteins were detected by western blot. * = NP detected from prior probing of the membrane before blotting for mCh. **(C)** RNP assembly monitored as in (B) except PB2 (FLAG) was targeted for immunoprecipitation with untagged PB2 as the negative control. **(D)** Polymerase activity assays with vNA-GFP were performed in the presence of mCh, MECR M77L-mCh, or cMECR-mCh. Fluorescence microscopy images taken at 24 h posttransfection and GFP-positive cells were enumerated by ImageJ. Mean ± SD of $n = 3$. Two-way ANOVA with post hoc Dunnett multiple comparisons test; ***, $P < 0.001$; **, $P < 0.01$; ns, not significant. **(E)** Phylogenetic maximum-likelihood analysis of MECR amino acid sequences. Δ, sequences lacking conserved cMECR start codon (Met77, human numbering). Circles indicate presence (filled circles) or absence (empty circles) of annotated cMECR transcripts. NCBI RefSeq IDs listed in S4 Table. **(F)** MECR KO cells were complemented with the MECR homolog from *Saccharomyces cerevisiae* Etr1 then transduced with cMECR. Cells were subsequently infected with WSN NLuc (MOI, 0.05; 24 h), and viral titers were measured in supernatants

and compared to WT. Mean ± SD of $n$ = 4. Unpaired two-tailed $t$ test; *, $P < 0.05$; **, $P < 0.01$. Below, mtFAS rescue was confirmed by detecting Lp-DLAT via western blotting whole cell lysates. Individual quantitative observations that underlie the data summarized here can be located under the Supporting information file as S1 Data. Uncropped images can be found in the Supporting information file as S1 Raw Images. CBB, Coomassie brilliant blue; cMECR, cytoplasmic MECR; Lp-DLAT, lipoylated dihydrolipoamide acetyltransferase; Lp-DLST, lipoylated dihydrolipoamide S-succinyltransferase; KO, knockout; mCh, mCherry; MECR, mitochondrial enoyl CoA-reductase; mtFAS, mitochondrial fatty acid synthesis; NLuc, nanoluciferase; NP, nucleoprotein; RNP, ribonucleoprotein complex; WT, wild-type.

reduced viral replication compared to WT cells (**Fig 6F**). Etr1 functioned in human cells to restore lipoic acid synthesis indicated by the return of lipoylated DLST in the complemented cells. Furthermore, Etr1 expression also increased virus growth to greater than WT levels, possibly due to the absence of cMECR. We then used the Etr1-complemented genetic background to demonstrate that expression of cMECR alone impairs virus growth (**Fig 6F**). These results were confirmed in two independent knockout cell lines (**S6B Fig**). The genetic complementation approach also demonstrated that the anti-influenza activity of cMECR does not require the presence of full-length MECR, which is important considering MECR is known to homodimerize [54]. Collectively, our results decouple the canonical role of MECR from the newly described antiviral activity of cMECR.

## Discussion

Zoonotic influenza viruses must gain a foothold to replicate and then adapt to their new human hosts. Here we used ICC-MS to identify human proteins that interact with influenza polymerases containing avian- or human-signature PB2 during infection. Network simulations and siRNA screening highlighted two key interactors, hnRNP UL1 and cMECR. We showed that hnRNP UL1 supports viral replication and likely acts as part of a pathway commonly exploited by viruses to aid mRNA splicing and transport. cMECR, by contrast, exerted antiviral activity by inhibiting RNP assembly and thus suppressing viral gene expression and replication. MECR has a well-defined role in mtFAS, yet this pathway was not identified in our network analyses and mechanistic studies showed that the antiviral activity was independent of mtFAS. MECR and mtFAS were generically important for viral replication, in as much as they were necessary for proper mitochondrial metabolism and cell health. Instead, we suggest that the antiviral activity derives from cMECR, an alternative splice variant that lacks the mitochondrial targeting sequence and is cytoplasmic. Using the yeast homolog Etr1 to supply the metabolic functions of MECR, we confirmed that antiviral activity is independent of mtFAS and reconstituted by ectopically expressing cMECR. Thus, our results support a model where a cryptic antiviral protein has been embedded within a key metabolic enzyme, possibly making it less susceptible to viral antagonism.

hnRNP family members, of which there are 37 in humans, broadly regulate host nucleic acid processes including chromatin organization, DNA damage repair, pre-mRNA processing and splicing, and mRNA nuclear export and subcellular transport [55,56]. As a consequence, hnRNPs are frequent cofactors for viral replication, including hnRNP UL1 we characterized here [57]. hnRNP UL1 contains RGG domains that can be methylated and are involved in RNA binding. However, neither of these activities were required for interaction with the viral polymerase (**Fig 3E**). hnRNP UL1 also binds to NXF1, the major cellular factor involved in nuclear export of viral mRNAs [33,39,58]. We speculate that hnRNP UL1 may enhance viral replication by bridging interactions between transcribing viral polymerases and NXF1 to facilitate mRNA export.

Mitochondria are a major subcellular location where host–pathogen conflicts play out: Cellular interferon (IFN) pathways use the mitochondrial membrane to assemble signaling hubs

containing mitochondrial antiviral-signaling protein (MAVS); mitochondrial nucleic acids can be unshielded to trigger the type I IFN response; and miRNAs that regulate IFN-stimulated genes are embedded in mitochondrial genes [59–62]. However, antiviral activity of *MECR* does not require innate sensing pathways that converge at the mitochondria as knockdown of *MECR* in A549 MAVS, RIG-I, or PKR knockout cells still alleviates its antiviral activity (**S6C Fig**). Instead, our data uncover another conserved mechanism where the metabolic enzyme MECR uses mRNA alternative splicing, and possibly leaky scanning, to moonlight as the antiviral protein cMECR. cMECR limits influenza A and B viruses, but not pandemic 2009 H1N1 (CA04) (**Fig 2E**). WSN and B/Bris are human origin strains, whereas the PB2 gene from CA04 is avian-like [63]. Human-derived PB2 genes localize to the mitochondrial matrix due to a mitochondrial targeting sequence in the N-terminus; the targeting sequences are disrupted in avian PB2 proteins by an N9D polymorphism [49,64,65]. Whether this contributes to differences in sensitivity to cMECR remains to be explored.

Phylogenetic analysis revealed that both MECR and the M77 that initiates cMECR are highly conserved in typical influenza hosts, including humans, birds, pigs, and horses. It is not yet known whether the absence of annotated cMECR transcripts in some species is due to bona fide differences in splicing or failed prediction algorithms. Among this conservation, the loss of M77 and potentially cMECR in *Salmo* species is especially interesting (**Fig 6E**; human amino acid numbering). *Salmo* species are hosts for infectious salmon anemia virus, an *orthomyxovirus* closely related to influenza virus. Analysis of splicing events in RNA-seq data shows that *Salmo* do not appear to utilize the alternative splice acceptor site that could create cMECR. Nonetheless, they may still express a cMECR-like protein by initiating translation at a different start codon, possibly the conserved M89 of MECR. This would require a distinct mechanism, such as leaky scanning, rather than the alternative splicing used for cMECR production in other vertebrates. *Salmo* do vary splicing between exons 1 and 2 by utilizing an alternative splice donor to create a splice isoform that encodes 26 amino acids not found in humans. It will be important to determine if *Salmo MECR* encodes a protein with antiviral activity, providing information on the antiviral function of MECR in different species and illuminating how conditions that alter splicing patterns or translation initiation sites stimulate or repress cMECR production.

A limitation of our study is the inability to uniquely detect endogenous cMECR and, as a consequence, to test cMECR function when natively expressed. cMECR is a minor variant and differs from MECR only by the absence of the first 76 amino acids. Smaller protein products were detected by blotting with antibodies targeting MECR (**Fig 4E**), but these were nonspecific as they were also present in both of our *MECR*$^{-/-}$ cell lines. Even if these were specifically detected, it would be challenging to differentiate whether these smaller fragments arose from translation of cMECR or from proteolysis of full-length MECR. We have used bioinformatic analysis, splicing assays and genetic approaches to address this limitation. RNA-seq identified splice isoforms that code for cMECR, and ribosome profiling provided support for translation initiation at the cMECR start codon (**Figs 5A and S6C**). Reporter assays showed that splicing of a model MECR transcript enabled initiation at the cMECR start codon (**Fig 5B**). Finally, ectopically expressing cMECR, but not MECR, was sufficient to confer an antiviral phenotype (**Fig 6A and 6F**). Nonetheless, the full antiviral activity of endogenous cMECR remains to be clearly demonstrated.

Viruses often exploit conserved essential host genes whose critical role for the cell limits mutational escape. Conversely, host antiviral genes are often nonessential and can undergo mutation or regulated expression to thwart infection and limit self-damage. Our study provides an example of a gene that is both essential and antiviral. Embedding the antiviral cMECR within the essential metabolic enzyme MECR establishes a Corneillian dilemma for influenza

virus. Viral antagonism of cMECR would require discrete targeting of the cytoplasmic isoform and could have the off-target effect of disabling MECR and mtFAS, a metabolic process important for cell viability and influenza virus output. Conversely, leaving MECR and mtFAS intact would ensure cell health, while the virus would remain vulnerable to cMECR. We suggest that the host strategy of encoding an alternatively spliced antiviral protein that shares structure with an energetically important protein may represent a failsafe for the host, forcing the virus to make the difficult "decision" of antiviral antagonism or maximum biosynthetic power.

## Methods

### Viruses, cells, plasmids, antibodies

Influenza viruses and plasmids were derived from A/WSN/33 (H1N1; WSN), A/green-winged teal/Ohio/175/1986 (H2N1; S009), A/California/04/2009 (H1N1; CA04), and B/Brisbane/60/2008 (B-Victoria lineage; B/Bris) [18,28,66,67]. Recombinant virus was rescued by transfecting cocultures of 293T and MDCK cells with pTM ΔRNP encoding WSN vRNA segments HA, NA, M, and NS and the bidirectional pBD plasmids encoding vRNA and mRNA for PB1, PA, NP, and the indicated PB2 mutants [18,68]. WSN PB2-FLAG and WSN-PB2-627E-FLAG viruses were previously described [35,69]. S009 PB2-FLAG and S009 PB2-627K-FLAG viruses contain NP and polymerase genes from S009 and the remaining segments from WSN and were constructed as previously described [18,70]. Nanoluciferase (NLuc)-expressing viruses were engineered to colinearly express PA, a 2A cleavage site, and NLuc (PASTN, referred to as WSN NLuc) on the third viral segment [71]. CA04 NLuc and B/Bris NLuc were generated similarly [28,67]. Viral stocks were amplified on MDBK cells and titered by plaque assay on MDCK cells. Influenza virus infections were performed by inoculating cells with stocks diluted in virus growth media (VGM; DMEM supplemented with penicillin/streptomycin, 25 mM HEPES, 0.3% BSA, and 0.25 to 0.5 μg/ml TPCK-trypsin).

VSV-G-pseudotyped lentivirus was prepared by transfecting 293T cells with pMD2.G (Addgene 12259), pLX304 (Addgene 25890) encoding HNRNPUL-1-V5 or MECR-V5; HNRNPUL1-V5 and MECR-V5) and psPAX2 (Addgene 12260) or pMD2.G, pQCXIP encoding Etr1-HA (Clontech) and pCIG-B [72]. Resultant viruses were used to transduce A549 cells. Cells were selected with blasticidin or puromycin to obtain stable expressing lines.

Mammalian cells were grown in DMEM supplemented with 10% FBS: 293T, ♀; A549, ♂; MDBK, ♂; MDCK, ♀. Innate sensing A549 knockout cells were a kind gift from C. McCormick and described previously [73]. All cells were maintained at 37°C, 5% $CO_2$. Cell stocks were routinely tested for mycoplasma (MycoAlert, Lonza). Human cell lines were authenticated by STR analysis (University of Arizona Genetics Core).

A549 MECR knockout cells were generated using CRISPR-Cas9 with a single guide RNA targeting exon 3 of *MECR* (GTTGCACAGGTGGTAGCGGTGGG designed by crispr.mit.edu). Annealed tracrRNA/gRNA was complexed with Cas9 (Alt-R; IDT), and RNPs were electroporated (Lonza) into cells following manufacturer's instructions. Bulk population (42% editing efficiency by Sanger sequencing and ICE analysis; [74]) was single cell sorted into 96-well plates by FACS 2 d post-nucleofection. Clones were screened by targeted next-generation sequencing of genomic DNA (Genome Engineering & iPSC Center, Washington University in St. Louis). Biallelic knockout cells were verified by western blot.

Cell proliferation assays were conducted over 3 d using CellTiter 96 AQ reagent (Promega) at 24 or 72 h after seeding, incubating 2 h, and measuring absorbance. Background absorbance at 630 nm was subtracted from 490 nm. Measurements were performed in technical triplicate and activity from 72 h subtracted from 24 h to determine growth.

pENTR HNRNPUL1 was generated using Gibson assembly from pDONR HNRNPUL1 obtained from DNASU.org (HsCD00719283). RNA binding mutants of HNRNPUL1 were generated by synthesizing mutant RGG boxes (nucleotides encoding amino acids 612–658; IDT) where all arginines were replaced with serine (SGG) or lysine (KGG) and introduced using Gibson assembly [41,42]. pDONR MECR was acquired from DNASU (HsCD00399762). pDONR cMECR was generated using inverse PCR to delete nucleotides encoding the first 76 amino acids of MECR. HNRNPUL1 and MECR were recombined into V5-tagged mammalian expression constructs by Gateway cloning into pcDNA6.2 (Invitrogen) and pLX304 (Addgene 25890). Codon optimized MECR was synthesized (IDT) containing an M77L mutation and cloned into pcDNA3 fused to a GGSGG (5GS) linker and mCh coding sequence on the 3′ end. Codon optimized cMECR was generated by inverse PCR using codon-optimized MECR as a template. pQCXIP (Qiagen) Etr1-HA was generated by Gibson assembly using pDONR Etr1 (DNASU; ScCD00009122). pcDNA3 FLAG-TPR was acquired from Addgene (60882). pcDNA6.2 NXF1 was generated by Gateway cloning pENTR NXF1 (DNASU; HsCD00514182).

Antibodies used for blotting include monoclonal anti-FLAG clones 1/27 and 1/54 made in-house at Roche, anti-FLAG M2-HRP (Sigma A8592), polyclonal anti-MECR (Proteintech 14932-1-AP or Atlas Antibodies HPA028740), polyclonal anti-hnRNP UL1 (Proteintech 10578-1-AP), polyclonal anti-V5 (Bethyl Labs A190-120A), monoclonal anti-V5-HRP (clone V5-10, Sigma V2260), polyclonal anti-PB1 [18]; monoclonal anti-tubulin (clone DM1A, Sigma T6199); polyclonal anti-lipoic acid (Calbiochem 437695); monoclonal anti-HA-HRP (clone 3F10, Sigma 12013819001), polyclonal anti-rabbit-HRP (Sigma A0545); and rabbit IgG (2729, Cell Signaling Technology). Antibodies used for immunofluorescence include mouse mono-clonal anti-FLAG (Sigma F1804), rabbit polyclonal anti-V5 (Bethyl Labs A190-120A), goat anti-mouse Alexa Fluor 594 (Invitrogen A-11032), and goat anti-rabbit Alexa Fluor 488 (Invitrogen A-11008).

## Immunocompetitive capture (ICC)

ICC was performed as previously published [17] with the following modifications. Anti-FLAG clone 1/27 was coupled to Affi-Gel 10 resin following the manufacturer's instructions (Bio-Rad). A549 cells were inoculated at an MOI of 0.2 with S009 PB2-FLAG or S009 PB2-627K-FLAG in a 10-cm dish, 5 dishes per replicate, 3 biological replicates. Infections were allowed to proceed for 24 h. Cells were combined and lysed in 1 ml co-IP buffer (50 mM Tris (pH 7.4), 150 mM NaCl, 0.5% NP-40, 1X cOmplete protease inhibitor (Roche)) and divided into equivalent fractions for ICC. Lysates were incubated by rocking for 3 h at 4˚C with free competing antibody (anti-FLAG 1/27) where applicable, then immunoprecipitated for 16 h with 1/27 antibody coupled to Affi-Gel 10 resin. After immunoprecipitation, samples were washed four times with co-IP buffer and eluted in Laemmli buffer at 70˚C for 10 min. Samples were then transferred to new tubes and boiled for 10 min with 0.1 M DTT. Approximately 10% of the immunoprecipitates were separated by SDS-PAGE and analyzed by western blot-ting using 1/54 anti-FLAG antibody or silver stained.

## Mass spectrometry (MS)

The remaining 90% of the ICC sample was separated on a 4% to 20% Tris-Glycine SDS-PAGE gel and stained with Coomassie blue. Lanes were cut from the gel and processed for in-gel digestion. Samples were analyzed with a nanoflow Easy-nLC 1000 system (Proxeon) connected to an Orbitrap Fusion Tribrid mass spectrometer and equipped with an Easy-spray source (Thermo Fisher Scientific). Samples were resuspended in LC–MS buffer (5% formic acid/2%

acetonitrile), concentrated on an Acclaim PepMap C18 trapping column (75 μm × 20 mm, 5 μm particle size), and peptides separated on an Acclaim PepMap C18 EASY-spray column (75 μm × 500 mm, 2 μm particle size) heated at 45°C using the following gradient at 300 nL/ min: 7% to 50% B in 45 min, 50% to 80% B in 2 min, 80% B for 13 min (buffer A: 0.1% formic acid; buffer B: 0.1% formic acid/acetonitrile). The instrument was set to collect Orbitrap MS1 scans over a mass range from $m/z$ 300 to 1,500 using quadrupole isolation, a resolution of 120,000 (at $m/z$ 200), an automatic gain control (AGC) target value of $2 \times 10^5$, and a maximum injection time (IT) of 100 ms. Using data-dependent acquisition (DDA) with a cycle time of 3 s between two MS1 scans, the most intense precursor ions with a minimum intensity of $5 \times 10^3$, were monoisotopically selected for high-energy collision dissociation (HCD) using a quadrupole isolation window of 0.7 Th, AGC target of $1 \times 10^4$, maximum IT of 35 ms, collision energy of 30%, and ion trap readout with rapid scan rate. Charge states between 2 and 6 and only one per precursor was selected for MS2. Already interrogated precursor ions were dynamically excluded for 20 s using a ±10-ppm mass tolerance.

MS raw files were processed for label-free quantification using Progenesis QI 2.1 (Nonlinear Dynamics), and ions m/z values were aligned to compensate for drifts in retention time between runs (maximum charge state set at +5). Peptides and proteins were identified by searching data with Mascot Server 2.5.1 (Matrix Science) together with the UniProt human (May 2016 release, 20,201 sequences) and the influenza S009 viral (12 sequences) protein databases. Searches used trypsin/P as an enzyme, a maximum of two missed cleavage sites, and 10 ppm and 0.5 Da as the precursor and fragment ion tolerances, respectively. Carbamidomethylated cysteines (+57.02146 Da) were set as static while oxidized methionines (+15.99492 Da) were set as dynamic modifications. The specFDR was restricted to 1% by performing a target-decoy search using a concatenated decoy database. Peptide extracted ion chromatograms (EICs) were used to determine peptide amounts. Data normalization was performed in Progenesis by applying a scalar multiple to each feature abundance measurement with the assumption that most peptide ions do not change in abundance (similar abundance distributions globally.

Data were quality controlled by assessing sample distribution and performing a principal component analysis. One outlier sample (Dose 0) was removed from the PB2-627K experiment, based on Mahalanobis distance of the first 3 principal components [75]. Specific interacting proteins should decrease in relative abundance with increased concentration of the free competitor antibody; these displaced proteins are determined as previously described [76]. Briefly, a linear model is fit on the $\log_2$-transformed relative abundance values for each protein with the free competitor compound concentration. Then, monotonic contrasts are used to compare the protein abundance values above and below each concentration point [77,78]. The maximum t-statistic from each series is determined with a moderated $t$ test [79]. Model fitting and post hoc contrast tests are performed with limma [80]. Significance of the displacement is assessed using permutation tests, where concentration labels were permuted 1,000 times based on the step-down minP algorithm [81] modified for one-sided tests, and adjusted for multiple testing [82]. Proteins with adjusted $p$-values below 5% are considered specific binders. Computations were performed in R [83].

## Network analysis

Network analysis was formulated as a minimum-cost flow optimization problem inspired by ResponseNet but customized for our application [84]. Human protein–protein interactions were obtained from the STRING database (version 10.5), using only interactions supported by experimental evidence [22]. PB2 interactors identified by ICC-MS were designated as source

nodes and the influenza host factors as target nodes in the STRING network. The host factors came from our prior genetic screens [28,29] and published RNA interference screens [23–27]. All human gene symbols were mapped to Ensembl protein identifiers. Proteins not present in this version of the STRING interaction network (i.e., PB2 interactors ABLIM1, EPDR1, and some influenza host factors) and TMEM106B were not included in the network analysis, leaving 23 source nodes and 2,179 target nodes.

The goal in the minimum-cost flow problem is to transport units of flow from a source node *S* in a network to a target node *T* [84]. *S* and *T* are special nodes that are added to the network and do not represent proteins. *S* has outgoing edges to all of the protein source nodes, the PB2 interactors. *T* has incoming edges from all the protein target nodes, the host factors. Flow can move from node to node through edges in the network. The total amount of flow to transport from *S* to *T* is fixed. However, each edge has its own cost associated with transporting a unit of flow over that edge and a capacity that limits how much flow it can transport. Flow was assigned to edges such that the total cost of transporting the fixed amount of flow is minimized. Additional constraints require that the total flow into a protein node in the network (i.e., all nodes except *S* and *T*) equals the total flow out of that node, the total flow from *S* equals the total flow into *T*, and the flow assigned to each edge is nonnegative and less than or equal to the edge's capacity. Solving the minimum-cost flow problem assigns how much flow each edge transports. The edges with positive flow compose a subnetwork, and in our application that subnetwork may comprise a predicted host influenza response pathway.

The standard minimum-cost flow problem was adjusted here to ensure that not all flow can be transported through one source or one target and a minimum number of sources and minimum number of targets that must transport positive flow was specified. The total flow to transport is then the product of these minimums. The network edges from *S* to the sources had capacity equal to the minimum number of targets. The network edges from the targets to *T* had capacity equal to the minimum number of sources. The protein–protein interaction edges had capacity equal to the total amount of flow, which did not constrain how much flow could be transported. The protein–protein interaction edges also had costs derived from STRING. The cost was one minus the STRING weight for the interaction, such that low confidence edges have higher costs. This construction guarantees that if a feasible solution exists, it will include at least as many sources and targets as requested. Furthermore, the solution will use the most confident protein interaction edges to transport that flow. If multiple equally good solutions exist, the solver selected one arbitrarily. If a solution could not be found, we reduced the minimum number of sources and targets. For the influenza network analysis, the minimum number of sources was set to 23 and the minimum number of targets to 200. This number of targets produced subnetworks that had multiple PB2 interactors in each connected component, which we also refer to as modules, as opposed to subnetworks that placed each source in its own connected component. The SimpleMinCostFlow solver from the ortools Python package (version 6.10.6025; https://developers.google.com/optimization) was used to solve the minimum-cost flow instances. Our Python code is available under the MIT license from https://github.com/gitter-lab/influenza-pb2 and archived on Zenodo at https://doi.org/10.5281/zenodo.7342881.

Flow simulations can be biased due to nodes forming spurious links in order to reach large multipartner targets. Two control analyses were conducted to control for this possibility and assess the significance of the protein interaction subnetwork from the minimum-cost flow solution. Both controls solve the minimum-cost flow problem many times using randomized input data that are not relevant to influenza A virus. If a protein belongs to both the influenza virus subnetwork and these control subnetworks, it may have been selected due to properties of the STRING network rather than influenza relevance. We defined an empirical *P* value for

each node in the influenza subnetwork as the number of times that node appears in a control subnetwork divided by the number of control runs. We executed 1,000 control runs. The simulated control sampled 23 source nodes, the number of real PB2 interactors, uniformly at random from all nodes in the STRING network. It used the real influenza host factors as target nodes. Results demonstrated that most of the flow subnetworks containing ICC-MS hits were specific to those generated by PB2 interactors and not randomly sampled proteins (S2A Table). The second type of control uses an alternative virus, hepatitis C virus, as the input. The alternative control sampled 23 source nodes from the 1,864 human proteins that interact with the hepatitis C virus nonstructural protein 5A as sources [17]. It used hepatitis C host factors from a CRISPR screen as targets [85]. This simulation demonstrated that the connection of most PB2 interactors to important viral cofactor nodes was specific to the influenza virus-defined network, and not the HCV-defined network (S2B Table). Both controls confirm that our PB2 interactors connect to subnetworks relevant for influenza virus, and not general virus replication or generic hubs.

The node sizes in the influenza subnetwork visualizations were scaled to be proportional to the node's negative $\log_{10} P$ value from the simulated and alternative controls. If a node's empirical $P$ value was 0, we set it to 0.001 for this visualization. In addition, subnetwork regions were annotated with enriched GO terms using the gprofiler-official Python package (version 0.3.5) for gene set enrichment analysis on each connected component of the subnetwork [86]. Each connected component-GO term enrichment result was ranked by a score that incorporated the negative $\log_{10} P$ value of the enrichment, the depth of the GO term in the biological process ontology, and the fraction of nodes in the connected component annotated with the GO term. The combined score emphasizes more specific GO terms that have statistically significant enrichment and cover a large fraction of nodes. For each connected component, we assigned the GO term with the largest combined score that had not already been assigned to a different connected component. The influenza subnetworks were visualized with Graphviz [87].

## siRNA screening and infection experiments

A549 or 293T cells were reverse transfected with 25 nM siRNA (SMARTpool, Horizon) using siQuest or X2 (Mirus) in 96-well plates for 48 h. Cells were inoculated with PASTN virus at an MOI of 0.1 (single cycle infection, 8 h) or 0.05 (multicycle infection, 24 h). Single cycle monolayers were seeded in white bottom plates and read directly for NLuc activity (Promega) on a Synergy HT plate reader (BioTek). Multicycle experiments were seeded in clear bottom plates and observed for siRNA toxicity. Supernatants were collected and titrated by infecting MDCK cells (white bottom 96-well plates) for 1 h with 20 μl supernatant, washing twice with VGM, and incubating for 8 h. Luciferase activity was read as above. Single cycle infections were also performed with CA04 (MOI, 0.5) and B/Bris (MOI, 1) PASTN viruses as above. Virus gene expression (single cycle infection) and titer (multicycle infection) were normalized to nontargeting siRNA control. For knockdown during WT virus infection, A549 cells were forward transfected for 2 d with 25 nM siRNA in 24-well plates and infected with WSN (MOI, 0.01) for 24 h. Supernatants were titrated by plaque assay on MDCK cells.

For overexpression experiments, 293T cells were reverse transfected with hnRNP UL1, NXF1, or TPR using TransIT-2020 (Mirus) in 96-well plates for 24 h. Transfected cells were infected with PASTN virus at an MOI of 0.01 for 24 h, and supernatants titrated as above. A549 cells stably overexpressing hnRNP UL1, MECR or cMECR, or MECR knockout cells overexpressing Etr1-HA were clonally isolated and analyzed for V5 or HA-tagged protein expression, respectively. Etr1-complemented cells were then transduced with cMECR-containing lentiviruses and polyclonal blasticidin-resistant cells used for experimental analysis.

A549 overexpression cells were infected with PASTN virus in 96-well plates for 24 h at an MOI of 0.05. Supernatants were titrated as above.

Infection experiments analyzing the role of mtFAS were performed by treating A549 cells with siRNAs for 48 h in 96-well plates with 50 nM nontargeting control siRNA, 25 nM nontargeting control mixed with 25 nM MECR siRNA, 25 nM nontargeting control mixed with 25 nM *NDUFAB1* (ACP) siRNA, or 25 nM of MECR and ACP siRNAs, followed by multicycle analysis with WSN PASTN as described above. For drug treatment, A549 cells were seeded in 96-well plates and treated with DMSO or 50 μM C75 (Sigma C5940) for 24 h, infected with WSN PASTN (MOI, 0.05; 24 h) in the presence of DMSO or 50 μM C75, and titrated as above.

## Co-immunoprecipitations

Interactions between endogenous hnRNP UL1 and PB2 were tested by co-IP. A549 cells were infected with WSN PB2-FLAG (10 cm dish; MOI, 1; 18 h), lysed in co-IP buffer, lysates immunoprecipitated overnight with anti-FLAG resin (M2, Sigma), and coprecipitating hnRNP UL1 was detected by western blot. For V5 immunoprecipitations, mammalian expression plasmids encoding PB2-FLAG, PB1-FLAG, PA, and HNRNPUL1-V5 or MECR-V5 were forward transfected into 293T cells using PEI (PEI MAX, Polysciences; 6-well plates). Cells were lysed in co-IP buffer, immunoprecipitated with anti-V5 antibody, and coprecipitating viral proteins were detected by western blot. To test the effect of viral RNA on polymerase:hnRNP UL1 interactions, plasmids expressing vNA or NP were included where indicated. Cells were lysed 2 d posttransfection in co-IP buffer with or without 1 μl RNase A (Thermo Scientific). Interactions with WT MECR during infection were tested by forward transfecting 293T cells with mammalian expression plasmids encoding PB2-FLAG, PB1-FLAG, PA, and MECR-V5 or cMECR-V5 for 42 h and infecting with WSN (MOI 10; 6 h) followed by lysis. Clarified lysates were incubated with 1.5 μg polyclonal anti-V5 or IgG for 1 h and captured with protein A agarose resin (P2545, Sigma) for 30 min. Immunoprecipitates were recovered, washed four times with co-IP buffer, and eluted by boiling in Laemmli sample buffer. Interactions with codon-optimized MECR-mCh and cMECR-mCh fusions were tested by transfecting 293T cells with mammalian expression constructs using TransIT-X2 (Mirus) for 48 h. Where indicated, plasmids expressing PB2-FLAG, PB1-FLAG, PA, NP, and genomic RNA vNA-GFP were cotransfected. Cells were lysed in co-IP buffer, mCh fusion proteins were captured with RFP-Trap (Chromotek RTA-20) for 3 h at 4° C, and resin was washed three times with co-IP buffer prior to elution by boiling in Laemmli sample buffer. Samples were separated by SDS-PAGE and analyzed by western blotting. Chemiluminescent images were captured on an Odyssey Fc Imager and quantified using Image Studio v5.2.5 (LI-COR). At least two biological replicate experiments were performed.

For concurrent analysis of cMECR interactions and RNP assembly, TransIT-X2 was used to transfect 293T cells with plasmids expressing codon-optimized MECR-mCh, cMECR-mCh, or mCh alone, along with PB2-FLAG, PB1-FLAG, PA, and the genomic RNA vNA-GFP in the presence or absence of NP-V5. Cells were lysed with co-IP buffer 2 d posttransfection and divided in half to assess interactions with cMECR using RFP-Trap or RNP assembly using V5-Trap (Chromotek V5TA-20). Samples were rocked for 3 h at 4˚C. Immunoprecipitates were recovered, washed once with 500 mM NaCl co-IP buffer, then twice more with co-IP buffer, and finally eluted by boiling in Laemmli sample buffer. Samples were separated by SDS-PAGE and analyzed by western blotting.

Proximity ligation utilized the biotin ligase AirID [53]. Assays were performed by expressing polymerase proteins in 293T cells with free AirID or AirID fused to the C-terminus of codon-optimized cMECR-V5. Exogenous biotin [50 μM] (Sigma B4501) was added 18 h posttransfection. Lysates were prepared at 48 h and PB2 was immunoprecipitated, blotted, and

probed with streptavidin-HRP (LI-COR 925–32230) or PB1 antibodies. Input samples were additionally probed with PB2 and V5 antibodies.

## Fluorescence microscopy

For immunofluorescence, 293T cells were transfected in 48-well plates with plasmids expressing V5-tagged hnRNP UL1, MECR, or cMECR and 48 h later infected with WSN PB2-FLAG (MOI, 3; 8 h). A549 cells stably expressing hnRNP UL1 or MECR were seeded on coverslips in 12-well plates and infected with WSN PB2-FLAG (MOI, 0.5; 8 h). Monolayers were fixed with 4% paraformaldehyde in PBS for 10 min, quenched and permeabilized with 0.1 M glycine + 0.1% Triton X-100 in PBS for 5 min, and blocked with 3% BSA in PBS for 30 min at room temperature. Primary and secondary antibodies were sequentially incubated for 1 h each at room temperature at 1 µg/ml in blocking buffer. DAPI was added to 293T cells during secondary antibody incubation. Coverslips were mounted in medium containing DAPI (Vector Laboratories, H-1200). Images were captured using a 20X objective on an EVOS FL Auto (Thermo Fisher) and processed in Adobe Photoshop CC.

For live cell imaging, 10X or 20X images were captured on an EVOS FL Auto 1 d posttransfection. To visualize nuclei, Hoechst 33342 (Anaspec AS-83218) was added 10 minutes prior to capture. GFP-positive cells were enumerated using ImageJ by automating the following commands: threshold (300); binary > watershed; analyze particles (size 100 to 500 px).

## RNA-sequencing

RNA was isolated from A549 cells that were mock treated, IFN-β treated (250 U/ml for 8 h), or infected with influenza WSN (MOI 0.02 for 24 h) using TRIzol (Invitrogen). Biologic triplicate sample RNA was sent for library preparation and paired-end RNA-seq by Novogene (BioProject PRJNA667475). Sequences were trimmed with BBDuk in the BBMap suite and aligned to the human genome (hg38) with HISAT2 [88,89]. Splicing events in the MECR locus were visualized in IGV (sashimi plot function) to enumerate the exon-joining reads into the 5′ boundary of exon 2 [90].

## cMECR initiation site identification and splicing reporter

Ribosome profiling data were acquired from Lee and colleagues [51]. In that study, HEK293 cells were treated with 50 µM LTM for 37°C for 30 min to arrest ribosomes at start sites prior to the first elongation event. We trimmed ribosome protected fragments from Illumina HiSeq 2000 runs (SRR618772, LTM rep1; SRR618773, LTM rep2) with BBDuk in the BBMap suite and aligned to the human genome (hg38) with HISAT2 [88,89]. Reads mapping to the *MECR* locus exons 1 and 2 were visualized with IGV.

cMECR splice reporters were constructed by inserting portions of the *MECR* locus upstream of the cMECR-V5 ORF. The native intron is approximately 14,000 bp; therefore, a miniaturized version was used for the "Splice" reporter. Exactly 80 nt from the 3′ of exon 1 and 100 nt from the 5′ end of intron 1 were fused 100 nt upstream of the cMECR splice acceptor site, leading into the remainder of the cMECR UTR and the cMECR start codon (M77 in MECR). The "Splice+FLAG" reporter was then created by replacing the fourth uORF with a start site and coding sequence for a FLAG epitope that remained in frame with cMECR-V5.

## Phylogenetic analysis

MECR amino acid sequences were retrieved from NCBI (S4 Table). Phylogenetic analysis was performed using Influenza Research Database (now the Bacterial and Viral Bioinformatics

Resource Center, https://www.bv-brc.org) using PhyML options [91] to generate a Newick file. FigTree v1.4.4 (http://tree.bio.ed.ac.uk/software/figtree) was used for visualization.

## Statistical analysis

Assays were performed with three to six technical replicates and represent at least three independent biological replicates. Mean and SD or SEM were calculated. When data were normalized, error was propagated to each individual experimental condition. Statistical significance was determined for pairwise comparisons with a Student *t* test and for multiple comparisons an ANOVA and post hoc Dunnett or Sidak test. Correlation across data sets was determined via a two-tailed Pearson correlation coefficient. Values from statistical analyses of siRNA screens are reported (S3 Table). Statistical tests were performed using Prism (v 8.4.2, GraphPad).

## Supporting information

**S1 Fig. Validation of ICC-MS and network analysis. (A)** Relative protein abundance of PB2 (red), PA (cyan), and NP (yellow) in PB2 ICC-MS samples shows decreasing capture with increasing competition antibody. Data shown are in biological triplicate. (**B**) Fully annotated and zoomable version of the diagram in Fig 1D. Minimum-cost flow simulations connect top PB2 interactors identified by ICC-MS (red) to influenza host factors (gray) through novel host proteins (white). Modules comprising different PB2 interactors with enriched GO terms and *P* values indicated. Node sizes indicate empirical *P* values based on the control flow simulation. (**C**) Enlarged view of MECR-containing module. (**D**) Validation shows that flow simulation networks of protein–protein interactions reflect biochemical pathways that regulate viral replication. Flow simulation captured PKR (*EIF2AK2*) interactions with RNA binding proteins DHX30, IGF2BP2, and TARBP and linked RIG-I (*DDX58*) with TRIM14 through MAVS and also IFIT3, LGP2, and USP15. Additionally, mRNA nuclear export pathways were faithfully reconstructed with NXF1 connecting with NXT2 and CRM1 (*XPO1)* connecting with NXF3. Our networks linked EXOSC3 to the exosome components EXOSC4, EXOSC5, and EXOSC8 and with the NEXT accessory complex (*RBM7*), both of which are important for influenza transcription [4]. Individual quantitative observations that underlie the data summarized here can be located under the Supporting information file as S1 Data. ICC-MS, immunocompetitive capture-mass spectrometry; MAVS, mitochondrial antiviral-signaling protein; MECR, mitochondrial enoyl CoA-reductase; NP, nucleoprotein.
(EPS)

**S2 Fig. Functional analyses of top candidate PB2 interactors. (A)** Secondary screening of proteomic hits by siRNA treatment and reporter virus infection. After knockdown, 293T cells were infected with human (PB2-627K; MOI, 0.01) or avian-adapted (PB2-627E; MOI, 0.05) WSN NLuc virus for 24 h. Viral supernatants were titered and normalized to an NT control. Control NXF1 (gray) and validated proteins highlighted (hnRNP UL1, cyan; MECR, yellow). Data are mean ± SEM of *n* = 2–3 biological replicates. (**B**) Concordance of virus titer for PB2-627E vs. PB2-627K virus infections in siRNA-treated cells (from (**A**)). Statistical analysis performed with a two-tailed Pearson correlation coefficient. (**C**) Screening as in (**A**), except A549 cells were infected with viruses as above for a single cycle of infection (MOI, 0.1; 8 h) and virus gene expression in infected cells normalized as above. (**D**) Concordance of gene expression (data from (**C**), analyzed as in (**B**)). Individual quantitative observations that underlie the data summarized here can be located under the Supporting information file as S1 Data. hnRNP UL1, heterogenous nuclear ribonuclear protein U-like 1; MECR, mitochondrial enoyl CoA-

reductase; NLuc, nanoluciferase; NT, nontargeting.
(EPS)

**S3 Fig. Enhanced expression of nuclear mRNA export proteins increases virus production.** (**A**) Replication of WSN NLuc (MOI, 0.05; 24 h) was measured in 293T cells transfected with EV or with plasmids encoding the indicated RNA export proteins. Mean ± SD of $n = 3$. One-way ANOVA with post hoc Dunnett multiple comparisons test; ****, $P < 0.0001$. (**B**) Immunofluorescence microscopy of WT and mutant hnRNP UL1 expressed in 293T cells, scale bars, 20 μm. Individual quantitative observations that underlie the data summarized here can be located under the Supporting information file as S1 Data. EV, empty vector; hnRNP UL1, heterogenous nuclear ribonuclear protein U-like 1; NLuc, nanoluciferase; WT, wild-type.
(EPS)

**S4 Fig. Allelic identities of *MECR* knockout A549 cells.** Deep sequencing of clonal knockout cells. CRISPR gRNA sequence underlined, PAM dotted. In-frame amino acids indicated on top. KO-1 is homozygous and KO-2 has two distinct alleles.
(EPS)

**S5 Fig. Characterization of endogenous *MECR* transcripts and cMECR:polymerase interactions. (A-B)** RT-PCR analysis of poly-adenylated *MECR* transcripts in mock or WSN-infected A549 cells (MOI 1, 6 h). (**A**) Genomic representation of *MECR* and regions amplified by specific primer sets. Untranslated regions (UTRs) in dark gray boxes, exons in light gray boxes, and introns in lines. Diagram segments are not to scale; base pair (bp) lengths are indicated. (**B**) Amplification of transcripts containing exon 1 (red), a region spanning exons 1 and 2 (green), and the unique cMECR UTR through exon 3 (yellow). The primer pair MECR ex1 FOR/MECR ex2 REV amplifies MECR (205 bp, empty circle) and cMECR (332 bp, filled circle). cMECR is predicted to be variable lengths around 332 bp due to the use of multiple splice donor sites paired with the cMECR UTR splice acceptor site. Asterisk; nonspecific band. Size shown in bp derived from a DNA ladder. (**C**) Ribosomes initiate on the short splice isoform transcript in the UTR and at the cMECR start site. Cells were treated with LTM to identify initiation sites during ribosome profiling (data from [51]). RPFs and total RNA (RNA-Seq) were mapped to the MECR locus, showing initiation events at the predicted cMECR start codon and other sites in the short splice isoform encoding cMECR. Zoom on exon 2 shows coding potential of the short isoform with stop codons in red, start codons in green, and the cMECR initiation site with an arrow. (**D**) PB2-FLAG and V5-tagged MECR or cMECR were expressed in 293T cells with or without the other polymerase (PB1/PA) or vRNP (vNA/NP) components. Cells were lysed and immunoprecipitated with anti-V5 antibody or IgG controls. Proteins were detected by western blot. (**E**) Subcellular localization of mCh fusion constructs. Plasmids encoding codon-optimized MECR (M77L) or cMECR fused to mCh with a 5GS linker were transfected into 293T cells. Red fluorescence images were captured 24 h posttransfection. Scale bars; 20 μm. (**F**) cMECR-AirID biotinylates polymerase. Cells coexpressing PB2-FLAG-tagged polymerase and free AirID or the cMECR-AirID fusion were immunoprecipitated with anti-FLAG. Input and coprecipitating proteins were detected by western blotting with streptavidin-HRP or antibodies recognizing the indicated proteins. Uncropped images can be found in the Supporting information file as S1 Raw Images. bp, base pair; cMECR, cytoplasmic MECR; IgG, immunoglobulin G; LTM, lactimidomycin; mCh, mCherry; MECR, mitochondrial enoyl CoA-reductase; RPF, ribosome-protected fragment; RT-PCR, reverse transcription PCR; UTR, untranslated region.
(EPS)

**S6 Fig. cMECR binds to and shuts down polymerase activity. (A)** Representative images used for GFP-positive cell counting in Fig 6D. GFP-based polymerase activity assays were performed in 293T cells coexpressing mCh, MECR M77L-mCh, or cMECR-mCh. NP was omitted in the negative control (−NP). Top: 10X whole-field view of vNA-GFP fluorescence. Bottom: Zoomed view with underlying Hoechst and mCh fluorescence. Dotted box indicates zoomed section. Scale bars for both images are 200 μm. (**B**) Etr1-expressing *MECR* knockout cells were transduced with lentivirus expressing cMECR or an empty vector control. Cells were subsequently infected with WSN NLuc (MOI, 0.05; 24 h), and viral titers were measured in supernatants. Mean ± SD of *n* = 3. Two-way ANOVA with post hoc Dunnett multiple comparisons test to compare to untransduced cells; *, $P < 0.05$; **, $P < 0.01$. **(C)** MECR antiviral effect does not require innate immune signaling pathways. siRNA-treated WT or *PKR, RIG-I*, or *MAVS* knockout A549 cells were infected with WSN NLuc (MOI, 0.05). Viral titers were measured 24 h postinfection and normalized to NT controls in the corresponding knockout line. Mean ± SD of *n* = 3. Two-way ANOVA with post hoc Dunnett multiple comparisons test to compare to WT cells; ****, $P < 0.0001$; other conditions were not significant. Individual quantitative observations that underlie the data summarized here can be located under the Supporting information file as S1 Data. cMECR, cytoplasmic MECR; mCh, mCherry; MECR, mitochondrial enoyl CoA-reductase; NLuc, nanoluciferase; NP, nucleoprotein; NT, nontargeting; WT, wild-type.
(EPS)

**S1 Table. PB2 interactors identified by ICC-MS.** WT or K627E PB2 was purified from infected cell lysate via immunocompetitive capture. Specific interactors were identified by ICC-MS (tabs a-c, pilot experiment tab e) and cross-referenced to prior MS studies (tab d).
(XLSX)

**S2 Table. Statistical support for minimum-cost flow networks.** Influenza networks were used to interrogate PB2 interactors (tabs a and b) or proteins with known interactors important for IAV replication (PKR, RIG-I, CRM1, NXF1, EXOSC3, and UBR4; tabs c and d). PB2 interactors formed 12 distinct modules, each enriched for different cellular processes (tabs e and f).
(XLSX)

**S3 Table. Statistical analysis of influenza virus infection siRNA screen.** Results from two-way ANOVA followed by Fisher LSD test for data presented in Fig 2.
(PDF)

**S4 Table. Accession numbers for MECR homologs.** Sequences used to build phylogenetic tree in Fig 6E.
(PDF)

**S1 Raw Images. Uncropped blots for Figs 1B, 2C, 3A, 3C, 3D, 4D, 4E, 5B, 5D, 5E, 6B, 6C, 6F, S5B, S5D and S5F.** Areas of interest used in figures are outlined with a red box. Experimental approach, antibody (where relevant), detection strategy, and equipment are indicated for each image. Sizes are shown corresponding to molecular weight markers for proteins or 100 bp ladders for PCR products. Lanes not used in the figure are marked with "X."
(PDF)

**S1 Data. Quantitative data underlying each figure with a tab associated with each panel in Figs 1–4, 6, S1–S3 and S6.**
(XLSX)

## Acknowledgments

We thank members of the Mehle lab for critical reading of the manuscript. We thank Craig McCormick (Dalhousie University) and Nathan Sherer (University of Wisconsin-Madison) for sharing reagents.

## Author Contributions

**Conceptualization:** Steven F. Baker, Helene Meistermann, Manuel Tzouros, Mitchell P. Ledwith, Anthony Gitter, Angelique Augustin, Hassan Javanbakht, Andrew Mehle.

**Data curation:** Steven F. Baker, Helene Meistermann, Manuel Tzouros, Aaron Baker.

**Formal analysis:** Steven F. Baker, Helene Meistermann, Manuel Tzouros, Aaron Baker, Juliane Siebourg Polster, Mitchell P. Ledwith, Anthony Gitter, Angelique Augustin, Andrew Mehle.

**Funding acquisition:** Steven F. Baker, Mitchell P. Ledwith, Anthony Gitter, Hassan Javanbakht, Andrew Mehle.

**Investigation:** Steven F. Baker, Helene Meistermann, Manuel Tzouros, Aaron Baker, Mitchell P. Ledwith, Andrew Mehle.

**Methodology:** Steven F. Baker, Helene Meistermann, Manuel Tzouros, Aaron Baker, Sabrina Golling, Angelique Augustin, Andrew Mehle.

**Project administration:** Helene Meistermann, Andrew Mehle.

**Supervision:** Helene Meistermann, Anthony Gitter, Angelique Augustin, Hassan Javanbakht, Andrew Mehle.

**Visualization:** Steven F. Baker, Aaron Baker, Andrew Mehle.

**Writing – original draft:** Steven F. Baker, Aaron Baker, Anthony Gitter, Andrew Mehle.

**Writing – review & editing:** Steven F. Baker, Helene Meistermann, Manuel Tzouros, Aaron Baker, Mitchell P. Ledwith, Anthony Gitter, Angelique Augustin, Hassan Javanbakht, Andrew Mehle.

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
