## [Editor Report · Decision Letter 0]

8 Nov 2022

Dear Dr. Andy, 

Thank you for submitting your manuscript entitled "Alternative splicing liberates a cryptic cytoplasmic isoform of mitochondrial MECR that antagonizes influenza virus" for consideration as a Research Article by PLOS Biology, and for your patience.

Your manuscript has now been evaluated by the PLOS Biology editorial staff, as well as by an academic editor with relevant expertise, and I am writing to let you know that we would like to consider your manuscript for publication without further peer-review. The manuscript will need to be revised to tone down the statements related to the cMERC endogenous expression and viral restriction. 

However, before we can send you a decision letter with the specific points to revise, we need you to complete your submission by providing the metadata that is required for full assessment. To this end, please login to Editorial Manager where you will find the paper in the 'Submissions Needing Revisions' folder on your homepage. Please click 'Revise Submission' from the Action Links and complete all additional questions in the submission questionnaire.

Once your full submission is complete, your paper will undergo a series of checks. After your manuscript has passed the checks, we will send you the decision letter. To provide the metadata for your submission, please Login to Editorial Manager (https://www.editorialmanager.com/pbiology) within two working days, i.e. by Nov 10 2022 11:59PM.

Kind regards,

Paula

---

Senior Editor

PLOS Biology

---

## [Editor Report · Decision Letter 1]

9 Nov 2022

Dear Dr Mehle,

Thank you for your patience while your manuscript "Alternative splicing liberates a cryptic cytoplasmic isoform of mitochondrial MECR that antagonizes influenza virus" was peer-reviewed at PLOS Biology. It has now been evaluated by the PLOS Biology editors and an Academic Editor with relevant expertise.

Based on our Academic Editor's assessment of your work, the previous reviewers' reports and the revised manuscript, we are likely to accept this manuscript for publication, provided you clearly state and discuss the following limitations. As raised by reviewer #1, you never show that the endogenous cMECR is produced and responsible for inhibition of influenza virus and the evidence that cMECR restricts influenza viruses under physiological conditions is not compelling. Similarly, the concept of embedding an antiviral activity in an essential function is interesting but not fully convincing in this case. You show that siRNA silencing of MECR increases virus replication. Thus, the virus should still be able to counteract this antiviral mechanism by reducing MECR and cMECR expression levels or by specific targeting of cMECR in the cytoplasm. Although whether endogenous cMECR indeed restricts IAV in infected cells remains to be clearly demonstrated, we think that the manuscript contains a lot of suggestive evidence for an interesting inhibitory mechanism. Ideally, to address this concern, presenting direct evidence for endogenous cMECR expression at levels sufficient for inhibition would greatly enhance the significance of the study. However, you certainly already tried hard to achieve this without success during the revision process. Therefore, it is very important to modify the abstract and the rest of the manuscript accordingly to clearly state and discuss the limitations of the work.

Please also make sure to address the following data and other policy-related requests.

1. DATA POLICY:

A) Supplementary files (e.g., excel). Please ensure that all data files are uploaded as 'Supporting Information' and are invariably referred to (in the manuscript, figure legends, and the Description field when uploading your files) using the following format verbatim: S1 Data, S2 Data, etc. Multiple panels of a single or even several figures can be included as multiple sheets in one excel file that is saved using exactly the following convention: S1_Data.xlsx (using an underscore).

B) Deposition in a publicly available repository. Please also provide the accession code or a reviewer link so that we may view your data before publication.

Regardless of the method selected, please ensure that you provide the individual numerical values that underlie the summary data displayed in the following figure panels as they are essential for readers to assess your analysis and to reproduce it: Figures 1CD, 2ABCD, 3B, 4BCEF, 6ADEF, and Supplementary Figures S1AB, S2ABCD, S3A, S6BC.

**Please also ensure that figure legends in your manuscript include information on where the underlying data can be found, and ensure your supplemental data file/s has a legend.**

**Please ensure that your Data Statement in the submission system accurately describes where your data can be found.**

We require the original, uncropped and minimally adjusted images supporting all blot and gel results reported in an article's figures or Supporting Information files. We will require these files before a manuscript can be accepted so please prepare and upload them now. We require this for Figures 1B, 2C, 3ACD, 4DE, 5CDE, 6BCF, and Supplementary Figure S5BDF.

Please carefully read our guidelines for how to prepare and upload this data: https://journals.plos.org/plosbiology/s/figures#loc-blot-and-gel-reporting-requirements

We expect to receive your revised manuscript within two weeks. Please, let me know if you need more time. 

*Published Peer Review History*

*Press*

Sincerely,

Paula

---

Senior Editor,

pjaureguionieva@plos.org,

PLOS Biology

---

## [Editor Report · Decision Letter 2]

29 Nov 2022

Dear Dr. Mehle,

Thank you for the submission of your revised Research Article "Alternative splicing liberates a cryptic cytoplasmic isoform of mitochondrial MECR that antagonizes influenza virus" for publication in PLOS Biology. On behalf of my colleagues and the Academic Editor, Frank Kirchhoff, I am pleased to say that we can in principle accept your manuscript for publication, provided you address any remaining formatting and reporting issues. These will be detailed in an email you should receive within 2-3 business days from our colleagues in the journal operations team; no action is required from you until then. Please note that we will not be able to formally accept your manuscript and schedule it for publication until you have completed any requested changes.

PRESS

Sincerely, 

Paula

---

Senior Editor

PLOS Biology
